# A new genetic strategy for targeting microglia in development and disease

Gabriel L McKinsey[1]*, Carlos O Lizama[2], Amber E Keown-Lang[1], Abraham Niu[1], Nicolas Santander[1], Amara Larpthaveesarp[1], Elin Chee[1], Fernando F Gonzalez[1], Thomas D Arnold[1]*

[1]Department of Pediatrics, University of California San Francisco, San Francisco, United States; [2]Cardiovascular Research Institute, University of California San Francisco, San Francisco, United States

**Abstract** As the resident macrophages of the brain and spinal cord, microglia are crucial for the phagocytosis of infectious agents, apoptotic cells and synapses. During brain injury or infection, bone-marrow derived macrophages invade neural tissue, making it difficult to distinguish between invading macrophages and resident microglia. In addition to circulation-derived monocytes, other non-microglial central nervous system (CNS) macrophage subtypes include border-associated meningeal, perivascular and choroid plexus macrophages. Using immunofluorescent labeling, flow cytometry and Cre-dependent ribosomal immunoprecipitations, we describe *P2ry12-CreER*, a new tool for the genetic targeting of microglia. We use this new tool to track microglia during embryonic development and in the context of ischemic injury and neuroinflammation. Because of the specificity and robustness of microglial recombination with *P2ry12-CreER*, we believe that this new mouse line will be particularly useful for future studies of microglial function in development and disease.

*For correspondence:
gabriel.mckinsey@ucsf.edu (GLM);
thomas.arnold@ucsf.edu (TDA)

**Competing interests:** The authors declare that no competing interests exist.

## Introduction

CNS macrophages can be broadly separated into three types: microglia, choroid plexus macrophages, and border-associated macrophages (BAMs) of the dura, subdura/pia and the perivascular space (*Goldmann et al., 2016*; *Wolburg and Paulus, 2010*). Circulatory monocytes, which invade the CNS and differentiate into macrophages, are also present in the context of CNS injury and inflammation (*Shechter et al., 2009*). These different types of macrophages play fundamental roles in CNS development, homeostasis and disease, but their specific functions remain poorly defined. An important barrier to defining the subtype-specific functions of CNS macrophages has been a lack of genetic tools to specifically target these subpopulations, although there has been some progress in this regard in recent years (*Chappell-Maor et al., 2019*; *Werner et al., 2020*; *Wieghofer et al., 2015*).

Microglia, the resident macrophages of the neural parenchyma, regulate a wide variety of processes in the brain, from development and synapse remodeling, to inflammatory insult and antigen presentation. Fate-mapping and developmental analyses have revealed that microglia are derived from the embryonic yolk sac, unlike circulating monocytes, which are derived from the adult bone marrow. Although bone marrow derived macrophages can adopt some features of endogenous microglia, they are not able to fully recapitulate all of their properties, suggesting that there may be important functional differences between these two populations of cells (*Bennett et al., 2018*). The use of single-cell transcriptomic profiling has revealed new insights about microglial biology in recent years (*Hammond et al., 2019*; *Jordão et al., 2019*; *Van Hove et al., 2019*) suggesting a greater degree of heterogeneity among microglia than previously recognized. Recently developed microglial genetic labeling strategies, such as the mouse line *Cx3cr1-CreER* (*Goldmann et al., 2013*;

*Parkhurst et al., 2013*; *Yona et al., 2013*), have significant advantages compared to their predecessors, but still have a number of drawbacks. For instance, *Cx3cr1* is expressed in other immune cell types in the brain, including perivascular, choroid plexus and meningeal macrophages (*Goldmann et al., 2016*). Acute tamoxifen administration in *Cx3cr1-CreER* mice also labels circulating monocytes, which can infiltrate the CNS in the context of injury or disease, thus complicating interpretation. This can be circumvented by administering tamoxifen and then waiting for recombined circulating monocytes to be replaced by non-recombined monocytes from the bone marrow (~15–30 days), so that only resident macrophages (e.g. BAMs and microglia) are recombined. Finally, *Cx3cr1-EGFP, -Cre and -CreER* mouse lines are haploinsufficient for *Cx3cr1,* as they were designed such that the *EGFP*, *Cre* or *CreER* coding regions replace the endogenous *Cx3cr1* coding region. There is evidence that *Cx3cr1* haploinsufficiency affects microglial function (*Hickman et al., 2019*; *Lee et al., 2010*; *Rogers et al., 2011*), so these lines should be used with this caveat in mind.

Here, we describe a *P2ry12-CreER* knock-in mouse line that we have developed to specifically label microglia. P2RY12 is a nucleotide sensing metabotropic GPCR in the P2Y family of GPCRs that has an important role in the microglial 'sensome' (*Hickman et al., 2013*). P2RY12 has been shown to regulate the morphological changes that microglia display in response to tissue damage or inflammation (*Bernier et al., 2019*; *Haynes et al., 2006*). Here, we found that *P2ry12-CreER*, unlike the commonly used *Cx3cr1-CreER*, specifically labeled brain microglia and a subset of dural and choroid plexus macrophages, but not pial-associated or perivascular macrophages. During embryonic development, *P2ry12-CreER* recombination was robust in microglia, although we also observed recombination in a small subset of meningeal and perivascular macrophages, which we suggest may reflect cells that are in a 'transition' state. Using *P2ry12-CreER* with a conditionally expressed *Rpl22-HA* allele, we performed Cre-dependent ribosomal immunoprecipitations and transcriptional profiling of adult microglia. By comparing this dataset to existing ribosomal profiles of CNS macrophages and subtracting genes that are shared by both datasets, we identify a number of border-associated macrophage markers, including *Pf4*. Analysis of the *Pf4-Cre* mouse line shows that *Pf4-Cre* recombination robustly labels BAMs and not microglia. Finally, we show that *P2ry12-CreER* can be used to label microglia in middle cerebral artery occlusion (MCAO)-induced ischemic stroke, as well as experimental autoimmune encephalomyelitis (EAE), a model of multiple sclerosis.

## Results

### Generation of a P2ry12-CreER mouse line

To identify specific markers of brain microglia, we cross-referenced published reports of microglial markers (*Buttgereit et al., 2016*; *Gautier et al., 2012*; *Wieghofer et al., 2015*) with recently produced single cell sequencing data sets (Tabula Muris: https://tabula-muris.ds.czbiohub.org, Myeloid Cell Single Cell Seq database: https://myeloidsc.appspot.com). We analyzed several microglial markers (*HexB*, *P2ry12*, *Sall1*, *Tmem119*, *Trem2*, *Fcrls* and others) (*Supplementary file 1*), and compared their expression across cell types in the mouse body. Among these genes, *P2ry12* appeared to be the most restricted to brain myeloid cells.

To genetically label P2RY12+ microglia, we generated a *P2ry12-2A-CreER* knock-in mouse line (hereafter called *P2ry12-CreER*) (*Figure 1A*) using CRISPR-facilitated homologous recombination. The targeting construct was designed to preserve *P2ry12* expression, with the *P2ry12* stop codon replaced by a ribosome skipping P2A-fusion peptide coding sequence, followed by the coding sequence for *CreER*. We did not find any obvious health problems in either *P2ry12-CreER* heterozygous or homozygous mice, which had normal life span, breeding, size/weight, skin/hair, no overt neuromotor deficits, and no obvious bleeding abnormalities observed during tissue harvesting. We did see a decrease in *P2ry12* transcript levels in the brains of homozygous *P2ry12-CreER* knock-in mice, but this did not lead to differences in P2RY12 protein expression or noticeable changes in microglial morphology in gene targeted versus wild-type mice (*Figure 1—figure supplement 1A–C and E*).

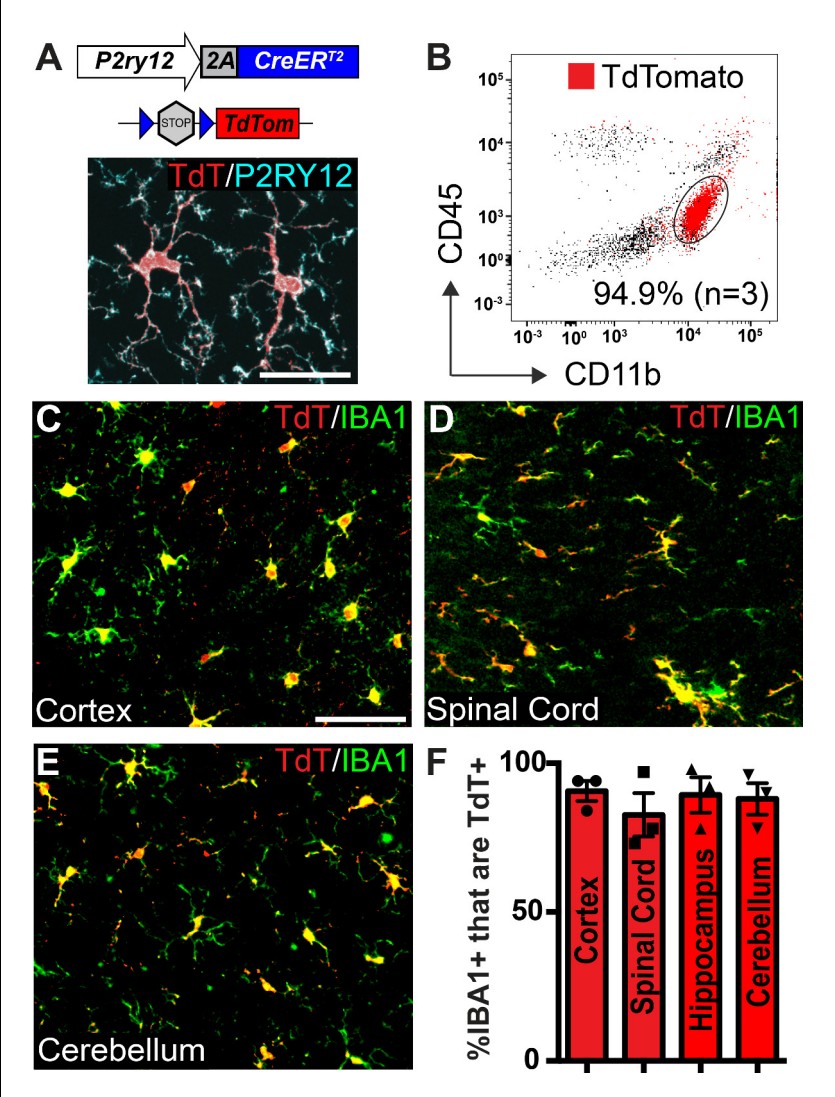

**Figure 1.** Microglia recombination by *P2ry12-CreER*. (**A**) Design of the *P2ry12* knock-in allele and a *P2ry12-CreER; Rosa26^{Ai14}* mouse brain section immunostained for P2RY12 (cyan) and TdTomato (red). (**B**) Flow cytometry analysis of recombined TdTomato+ microglia in a *P2ry12-CreER;Rosa26^{Ai14}* mouse. Microglia were pre-gated on forward scatter, isolation of single cells, and removal of dead cells. TdTomato+ cells are marked in red. 94.9 ± 2.75% of microglia (CD11b$^+$CD45$^{int}$ cells) were recombined by *P2ry12-CreER*. (**C-E**) Images of recombination in microglia of the cerebral cortex (**C**), spinal cord (**D**) cerebellum (**E**) in *P2ry12-CreER; Rosa26^{Ai14}* mice. Sections stained with pan-macrophage marker IBA1 (green). (**F**) Immunohistochemical quantification of recombination in the cerebral cortex, spinal cord, hippocampus and cerebellum. For B-E, n = 3 mice. Error bars in F = standard error of the mean (SEM). Scale bars = 20 µm (**A**), 50 µm (**C**).

The online version of this article includes the following source data and figure supplement(s) for figure 1:

**Source data 1.** Microglial recombination by *P2ry12-CreER*.
**Figure supplement 1.** qPCR and western blot analysis of *P2ry12* expression; background recombination of *P2ry12-CreER*.
**Figure supplement 2.** Flow cytometry gating strategy used for isolating microglia and TdT+ cells.

## P2ry12-CreER-mediated recombination efficiently and specifically labels microglia

To determine the efficiency and specificity of *P2ry12-CreER* recombination, we generated *P2ry12-CreER; Rosa26^{Rosa26Ai14}* reporter mice, which express TdTomato upon Cre-dependent recombination (*Madisen et al., 2010*). There was very low, but not zero recombination in the absence of

tamoxifen (.38±. 07% = of microglia,~1 in 263, in the cerebral cortex of *P2ry12-CreER; Rosa26^Ai14* heterozygotes, *Figure 1—figure supplement 1D*). We suspect that this non-tamoxifen recombination rate will likely be lower for other floxed alleles, given the sensitivity of the Rosa26^Ai14 reporter (*Álvarez-Aznar et al., 2020*). Following tamoxifen induction (3 injections of 150 μL 20 mg/mL tamoxifen in corn oil, delivered by oral gavage every other day) in adult mice we saw robust TdTomato expression in P2RY12$^+$ microglia (*Figure 1A*). Analysis by flow cytometry of microglial recombination in the brain revealed that 94.9 ± 3.4% of microglia (CD11b$^+$CD45$^{int}$ cells) expressed TdTomato (*Figure 1B*). Immunohistochemical analysis showed similarly high levels of recombination in CNS IBA1+ microglia (90 ± 3.0% in the cerebral cortex, 82.7 ± 7.25% in the spinal cord, 89.2 ± 6.1% in the hippocampus and 88.0 ± 5.4% in the cerebellum)(*Figure 1C–F*).

## P2ry12-CreER-mediated recombination in border-associated macrophages (BAMs)

To determine the specificity of *P2ry12-CreER*, we examined recombination in different subsets of border-associated macrophages (*Figure 2A*). We directed our initial analysis to LYVE1+, SMA-adjacent pial and perivascular macrophages. LYVE1 (lymphatic vessel endothelial hyaluronan receptor 1) is a membrane glycoprotein that marks the lymphatic vasculature, but is also expressed in meningeal and perivascular macrophages. Immunostaining thin coronal brain sections for LYVE1 and SMA (which marks mural smooth muscle cells around large brain vessels) revealed that *P2ry12-CreER; Rosa26^Ai14* recombination did not mark LYVE1+ perivascular macrophages (*Figure 2B*), unlike *Cx3CR1-CreER; Rosa26^Ai14* recombination (*Figure 2—figure supplement 1A*). Immunostaining for CD206 (MRC1), an additional marker of perivascular macrophages, also revealed no overlap with *P2ry12-CreER; Rosa26^Ai14* recombination (*Figure 2D*). Similarly, we saw no *P2ry12-CreER; Rosa26^Ai14* recombination in LYVE1+ macrophages of the pia (*Figure 2C*), unlike *Cx3cr1-CreER; Rosa26^Ai14* (*Figure 2—figure supplement 1B*). In the choroid plexus, *P2ry12-CreER* recombination marked 21.8 ± 3.7% of IBA1$^+$ cells, unlike *Cx3cr1-CreER*, which showed recombination in most choroid plexus IBA1$^+$ macrophages (*Figure 2E,J*, and *Figure 2—figure supplement 1C*). *P2ry12-CreER; Rosa26^Ai14* recombined cells appeared to be on the surface of the choroid plexus, consistent with a recently published study that described a microglia-like, *P2ry12*-expressing 'CP-epi' subpopulation of choroid plexus macrophages, which may be Kolmer's epiplexus cells (*Ling et al., 1998*; *Van Hove et al., 2019*).

To better examine meningeal recombination in *P2ry12-CreER; Rosa26^Ai14* mice, we performed whole-mount immunostaining of these tissues according to published methods (*Van Hove et al., 2019*). Labeling these preparations for two different BAM markers (LYVE1 and CD206) showed that *P2ry12-CreER; ROSA26^AI14* recombination occurred in a subset of macrophages in the dura mater (24.3 ± 2.6% of CD206+ and 25.0 ± 0.9% of LYVE1+ macrophages), but did not recombine macrophages of the pia mater or perivascular space (*Figure 2F–I* and *Figure 2—video 1*).

## P2ry12-CreER recombination efficiently labels embryonic microglia

In our previous studies, we noted that P2RY12 expression is developmentally regulated, increasing in the brain throughout embryonic and early post-natal development (*Arnold et al., 2019*). To determine whether *P2ry12-CreER* was capable of recombining embryonic macrophages, we induced pregnant dams carrying *P2ry12-CreER; Rosa26^Ai14* pups with a series of three tamoxifen doses at E13.5, E15.5, and E17.5 (*Figure 3A*). We collected these pups at E18.5 and analyzed their degree and pattern of recombination. Examination of these brains revealed robust recombination in IBA1$^+$ parenchymal microglia; all microglia examined were TdTomato+. IBA1+ recombined microglia were most densely concentrated in the subventricular zone of the cortex (SVZ) and lateral migratory stream of the amygdala (*Figure 3B–B'', C,F*). Compared to the adult choroid plexus, we observed a larger percentage of recombination in macrophages of the embryonic choroid plexus (41.9 ± 5.8% of IBA1+ cells in the embryos vs 21.8 ± 3.7% in the adult) (*Figure 3K*). We also observed a low level of recombination in the embryonic meninges (8.0 ± 3.1% of CD206+ cells, 12.9 ± 2.5% of LYVE1+ cells). In contrast to adult *P2ry12CreER; Rosa26^Ai14* mice, we observed recombination in macrophages apposed to nascent cerebral arteries. (*Figure 3H*). We did not observe any obvious recombination in the blood (embryos were not perfused prior to immersion fixation), vascular endothelium, or in other cell types in the brain at this age.

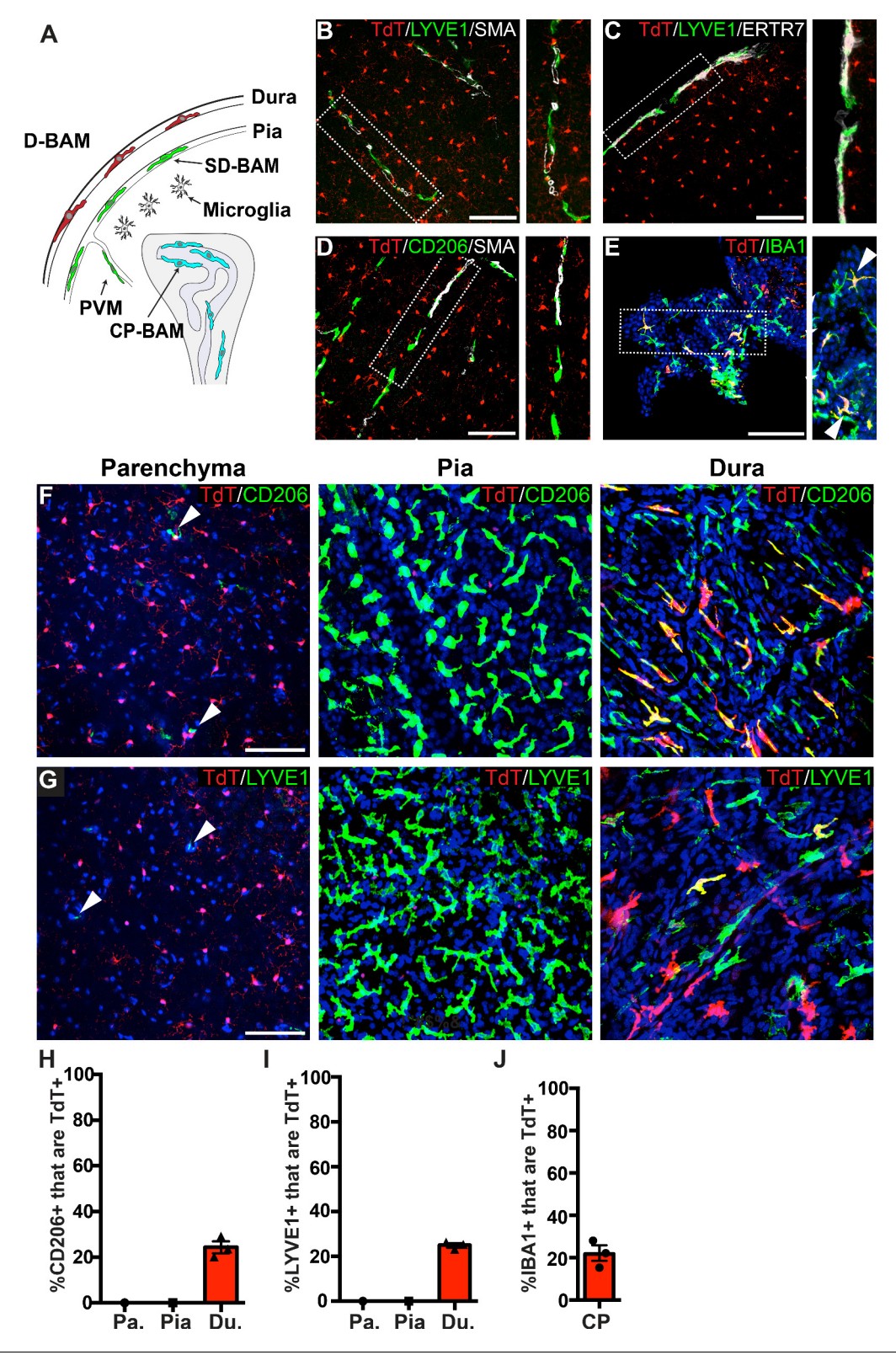

**Figure 2.** Analysis of *P2ry12-CreER* recombination in brain macrophage populations. (**A**) Cartoon depicting the subsets of brain macrophage populations that were analyzed in *P2ry12-CreER; Rosa26^{Ai14}* mice. D-BAMs, SD-BAMs, CP-BAMs = dural, subdural, choroid plexus border-associated macrophages. PVMs = perivascular macrophages. Adapted, with permission, from *Van Hove et al., 2019*. (**B**) Analysis of *P2ry12-CreER; Rosa26^{Ai14}* recombination in SMA-adjacent LYVE1+ perivascular macrophages. No recombination was seen in perivascular macrophages, unlike *Cx3CR1-CreER; Figure 2 continued on next page*

*Figure 2 continued*

*Rosa26^{Ai14}* mice (see *Figure 2—figure supplement 1*). (C) Analysis of recombination in LYVE1+ pial macrophages. ERTR7 expression delineates pial boundaries. (D) Analysis of recombination in SMA-adjacent CD206+ perivascular macrophages. (E) Analysis of recombination in IBA1+ macrophages of the choroid plexus. (F) Whole-mount cerebral cortex and dura, immunostained for CD206. (G) Whole-mount cerebral cortex and dura, immunostained for LYVE1. Weakly red cells in pial images are microglia in deeper focal planes. (H–I) Quantification of recombination in CD206+ (H) and LYVE1+ (I) macrophages in the parenchymal, pial and dural spaces. (J) Quantification of recombination in IBA1+ macrophages of the choroid plexus. Error bars = SEM. Scale bars = 100 μm (B–G).

The online version of this article includes the following video, source data, and figure supplement(s) for figure 2:

**Source data 1.** Analysis of P2ry12-CreER recombination in brain macrophage populations.
**Figure supplement 1.** Analysis of perivascular and pial recombination in *Cx3CR1-CreER; Rosa26^{Ai14}* mice.
**Figure 2—video 1.** Z-stack image series of *P2ry12-CreER; Rosa26^{Ai14}* cortical whole-mount immunostaining.
https://elifesciences.org/articles/54590#fig2video1

In our immunostains of recombined E18.5 embryonic brains with CD206, we also noticed a population of recombined TdT+; CD206+ cells in the embryonic hippocampus and cortical SVZ (*Figure 2G*). These double positive cells were most frequent at the junction of the developing hippocampus and the overlying meninges (arrowheads in *Figure 3G*), but were also found in the hippocampal and cortical SVZ. CD206 expression was strongest in the hippocampus in cells that were radially aligned relative to the overlying meninges (arrow in *Figure 3G*). With greater distance from the hippocampus, CD206 expression in the SVZ was less frequent and intense, possibly indicating active down-regulation of CD206 in infiltrating meningeal macrophages as they migrate through the hippocampal and cortical SVZ (see discussion).

## Expression of P2ry12-CreER in other tissues and blood

To determine the specificity of *P2ry12-CreER*-mediated recombination, we examined recombination in non-microglial cell types in the brain, circulation and in a variety of organs. In the brain, we saw no recombination in macroglia (oligodendrocytes and astrocytes), or neurons (*Figure 4A*). In the heart, intestine, lungs and spleen, TdTomato expression was limited to a small subset of CD206+ Cx3cr1-EGFP+ cells, suggesting that *P2ry12* expression marks a subset of macrophages in these tissues (*Figure 4B* and quantification in 4C). The kidney did not show any TdTomato expression (not shown). In the thymus, *P2ry12-CreER; Rosa26^{Ai14}* recombination partially overlapped with Cx3cr1-EGFP and CD206+. The most prominent non-CNS recombination was in the spleen, where recombination was moderate (27.5 ± 5.8% of Cx3cr1-EGFP+ cells of the marginal zone and white pulp) and primarily restricted to CD206+; Cx3cr1-EGFP+ cells of the marginal zone between the red and white pulp, an area that contains macrophages that surveil the bloodstream, although there were a small number of TdT+ cells in the white pulp as well (*Figure 4B*, *Figure 4—figure supplement 1A–B*; *Borges da Silva et al., 2015*). In the liver, *P2ry12-CreER; Rosa26^{Ai14}* recombination was excluded from perivascular SMA-adjacent Cx3CR1-EGFP cells, but was seen in Kupffer cells, in line with previous reports that Kupffer cells lack Cx3cr1 expression (*Figure 4—figure supplement 1C*; *Yona et al., 2013*). Unlike the liver, *P2ry12-CreER; Rosa26^{Ai14}* recombined cells in the lungs were close to SMA-demarcated airways and associated blood vessels (*Figure 4—figure supplement 1D*). In an analysis of *P2ry12-CreER; Rosa26^{Ai14}* recombination in embryonic tissue, we did not see any recombination in the embryonic liver or heart. In the intestine and heart, we saw low levels of recombination that partially overlapped with CD206 and IBA1 (*Figure 4—figure supplement 2*).

To determine whether *P2ry12-CreER* recombines circulating blood cells and platelets, we examined recombination in *P2ry12-CreER; Rosa26^{Ai14}* adult mice using blood smears and flow cytometry, one day after tamoxifen administration (same dosing as above). Blood smears of tamoxifen induced *P2ry12-CreER; Rosa26^{Ai14}; Cx3Cr1-EGFP* mice showed no recombination in circulating lymphocytes, but as an internal positive control, these mice did have numerous Cx3cr1-EGFP$^+$CD45$^+$CD11b$^+$ circulating monocytes (*Figure 4D*). In contrast, smears of *Cx3cr1-CreER; Rosa26^{Ai14}* mice showed TdTomato expression indicative of recombination in circulating monocytes. Flow cytometry of *P2ry12-CreER; Rosa26^{Ai14}; Cx3Cr1-EGFP* mice showed that while there were significant numbers of EGFP+ monocytes (34.93 ± 2.6% of CD45$^+$;CD11b$^+$ cells), there was a lack of TdTomato$^+$ circulating monocytes (.0175 ± 0.02% of CD45$^+$;CD11b$^+$ cells, 3 samples with 0 TdTomato$^+$ cells, 1 sample with 2 TdTomato$^+$ cells out of 2852 CD45$^+$;CD11b$^+$ monocytes). In comparison, *Cx3cr1-CreER;*

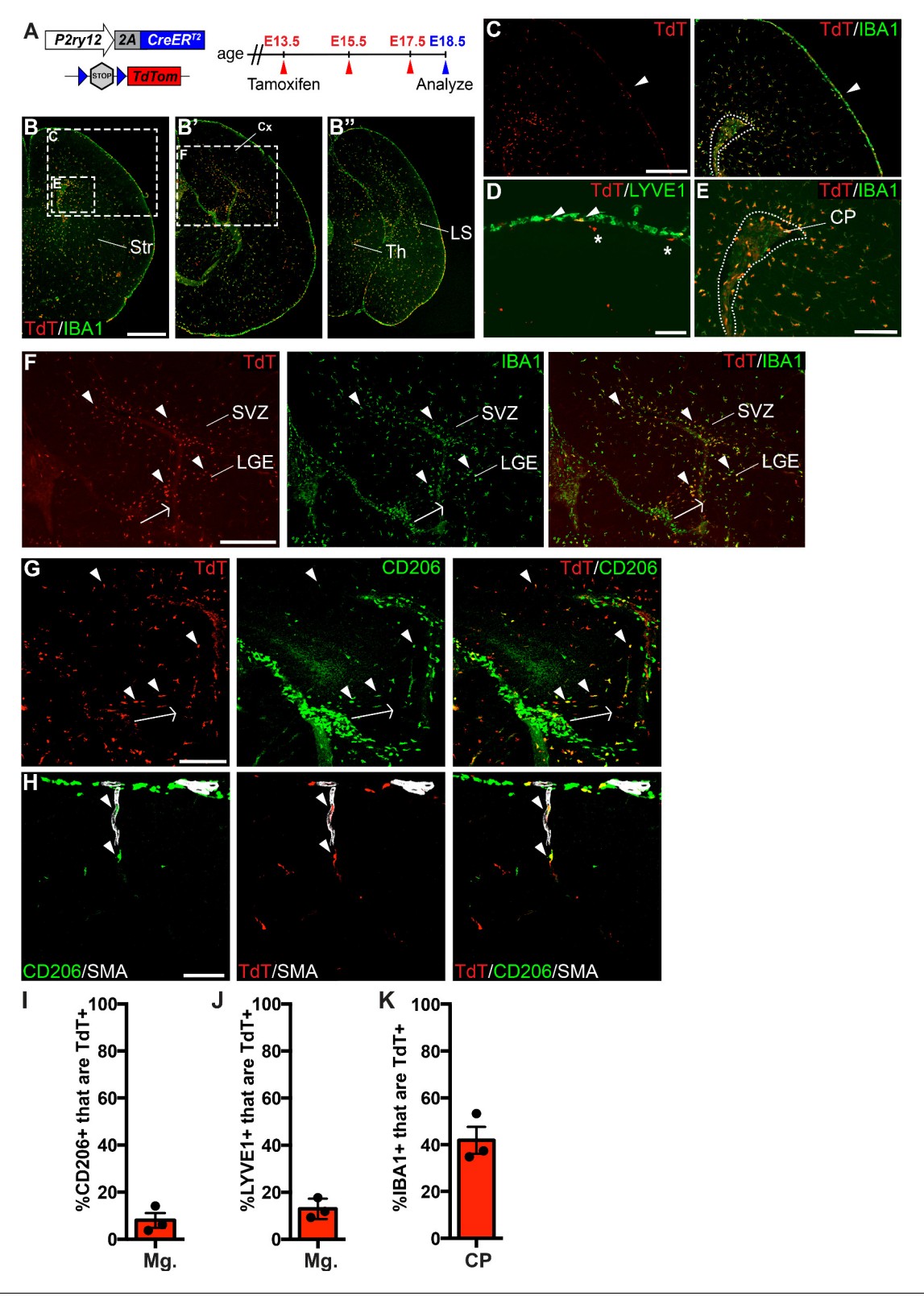

**Figure 3.** *P2ry12-CreER* recombination in the embryonic brain. (A) Diagram of *P2ry12-CreER; Rosa26$^{Ai14}$* embryonic tamoxifen induction regimen. Pregnant mice were induced with tamoxifen at E13.5, E15.5 and E17.5. Embryos were collected at E18.5. (B) Recombination in the E18.5 brain, shown in three images, arranged rostral (B) to caudal (B''). (C) *P2ry12-CreER; Rosa26$^{Ai14}$* recombination in the developing cerebral cortex. Recombination was seen primarily in the parenchyma of the brain, with low levels of recombination in the meninges (arrowhead). This is quantified in I and J. (D)

*Figure 3 continued on next page*

*Figure 3 continued*

Recombination in LYVE1+ cells of the embryonic meninges (arrowheads). Recombination in microglia is noted with asterisks. (E) Recombination in IBA1 + cells of the choroid plexus, quantified in K. (F) Recombination was most concentrated in IBA1+ cells of the hippocampal and cortical SVZ (arrowheads). (G) CD206 staining of recombined cells of the hippocampal SVZ and cortical SVZ. CD206 expression was lower, and coexpression with TdT less frequent, with greater distance from the hippocampus. (H) Recombination was observed in SMA-adjacent CD206+ perivascular macrophages at E18.5 (arrowheads). (I–J) Quantification of recombination in meningeal CD206+ (I) and LYVE1+ (J) macrophages. (K) Quantification of recombination in embryonic choroid plexus IBA1+ macrophages. Cx = Cortex, LGE = lateral ganglionic eminence, LS = Lateral migratory stream, Str = Striatum, Th = Thalamus, SVZ = Subventricular zone. Scale bars = 800 µm (B), 400 µm (C, F), 100 µm (D, G, H), 200 µm (E).

The online version of this article includes the following source data for figure 3:

**Source data 1.** P2ry12-CreER recombination in the embryonic brain.

*Rosa26^{Ai14}* samples showed recombination in 20.3 ± 1.8% of CD45$^+$;CD11b$^+$ monocytes (*Figure 4D*). Surprisingly, we observed no evidence of platelet recombination, using platelet factor 4 (*Pf4*)-*Cre*; *Rosa26^{Ai14}* mice (well known to recombine platelets) as a positive control (*Figure 4—figure supplement 1E*). These data indicate that while *P2ry12-CreER* recombination labels sparse subsets of resident macrophages in organs other than the brain, it does not label circulating blood cells or platelets.

## P2ry12-CreER-dependent ribosomal profiling enriches for transcripts of parenchymal microglia

We next studied the mRNA transcriptional profile of microglia using *P2ry12-CreER* mice. Classically, mRNA profiling of defined myeloid cell types requires preparation of single cell suspensions followed by cell sorting, RNA isolation and sequencing. These manipulations are inefficient and introduce bias, especially in macrophage populations, which are difficult to mechanically or enzymatically separate. Furthermore, microglia show significant transcriptional changes upon dissociation and sorting (*Haimon et al., 2018*) although some changes can be suppressed by treatment with the transcription inhibitor actinomycin D (ActD-Seq) (*Van Hove et al., 2019*). To avoid these pitfalls, we performed TRAP-Seq (translating ribosome affinity purification followed by mRNA sequencing) (*Sanz et al., 2009*) which allows for cell-specific isolation of mRNA without these manipulations. We generated *P2ry12-CreER*; *Rpl22-HA* mice, which, upon recombination, express a hemagglutanin (HA) epitope tagged RPL22 riboprotein that can be isolated by immunoprecipitation (*Figure 5A*). Upon tamoxifen-induced recombination in *P2ry12-CreER*; *Rpl22-HA* mice, we found robust labeling of microglia, as shown by immunofluorescent costaining for HA and IBA1 (*Figure 5B*). After qPCR validation for enrichment of microglial transcripts, libraries from *P2ry12-CreER*; *Rpl22* immunoprecipitations were generated and sequenced (workflow in 5C). Analysis of the transcripts enriched in *P2ry12-CreER*; *Rpl22-HA* libraries revealed significant enrichment for many known microglial markers, but did not show enrichment for markers of other cell types, such as neurons, oligodendrocytes and astrocytes (*Figure 5D*). Of note, we performed these experiments prior to learning of dural macrophage recombination by *P2ry12-CreER* mice (*Figure 2*), and did not include the dura in our immunoprecipitations.

The transcriptional profile of *P2ry12-CreER* recombined cells was quite similar to recently published profiles derived from *Cx3cr1-CreER*; *Rpl22-HA* immunoprecipitations (*Haimon et al., 2018*). By comparing the relative enrichment values for individual genes in our *P2ry12-CreER*; *Rpl22-HA* data (X axis, *Figure 5E*) to *Cx3cr1-CreER*; *Rpl22-HA* enrichment values for the same gene (Y axis, *Figure 5E*), we identified a number of enriched genes that were either common or unique to each dataset. Genes common to both datasets included macrophage markers such as *Spi1*, *Aif1 (which codes for IBA1)*, *Csf1r* and *Cd11b*. Genes specific for neurons, macroglia, vascular endothelium, and blood cells/platelets (not shown) were depleted in both datasets. In line with our histological and cell-sorting data, there were many genes enriched in *Cx3cr1-CreER*; *Rpl22-HA* IPs that were not enriched in *P2ry12-CreER*; *Rpl22-HA* IPs. Some of these genes have recently been identified in single-cell sequencing studies as BAM-specific markers (*Van Hove et al., 2019*), suggesting that our comparison differentiates genes expressed in parenchymal microglia from those expressed in non-parenchymal macrophages in the CNS (*Supplementary file 2*). This subtractive approach also identified a number of genes that were preferentially enriched in the *P2ry12-CreER*; *Rpl22-HA* population,

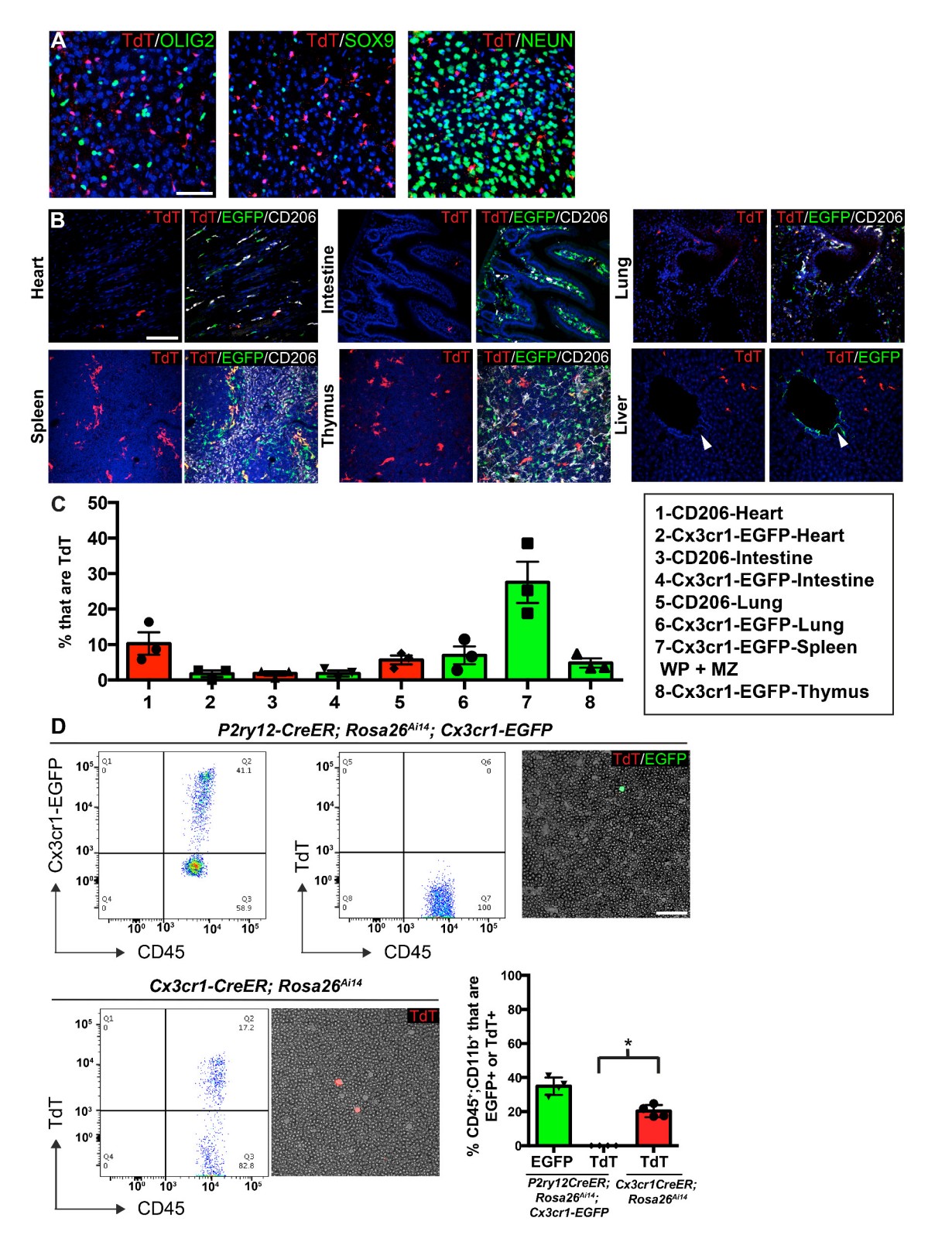

**Figure 4.** *P2ry12-CreER* recombination in non-microglial populations. (**A**) Analysis of *P2ry12-CreER; Rosa26^{Ai14}* recombination showed a lack of recombination in oligodendrocytes (OLIG2), astrocytes (SOX9) or neurons (NEUN). (**B**) Analysis of recombination and coexpression with Cx3cr1-EGFP and CD206 in the spleen, lungs, heart, thymus, intestine and liver of adult *P2ry12-CreER; Rosa26^{Ai14}; Cx3cr1-EGFP* mice. (**C**) Quantification of recombination overlap with Cx3CR1-EGFP (heart, intestine, lungs, spleen, thymus) or CD206 (heart, intestine, lungs). CD206 coexpression in the spleen

*Figure 4 continued on next page*

Figure 4 continued

and thymus was not analyzed due to density of CD206 expression. (D) Flow cytometry and blood smear recombination analysis revealed negligible recombination of blood cells in *P2ry12-CreER;Rosa26^{Ai14}* mice compared to *Cx3cr1-EGFP* (internal control), or to *Cx3cr1-CreER; Rosa26^{Ai14}* mice. *p=0.0015, Student's t-test. N = 4 mice for A. and D., N = 3 mice for B and C. Scale bars = 50 µm (A), 100 µm (B,D).
The online version of this article includes the following source data and figure supplement(s) for figure 4:

**Source data 1.** P2ry12-CreER recombination in non-microglial populations.
**Figure supplement 1.** *P2ry12-CreER* recombination in the spleen, liver, lungs and circulating platelets.
**Figure supplement 2.** *P2ry12-CreER* recombination in embryonic organs.

which represent potential microglial-specific markers. As expected, *P2ry12* was in this group, as were a number of other *P2Y* family members, such as *Gpr34*, *P2ry13* and *P2ry6*.

## Pf4-Cre recombination efficiently labels BAMs

One gene that significantly enriched in *Cx3cr1-CreER; Rpl22-HA* IPs but marked depletion in *P2ry12-CreER; Rpl22-HA* IPs was the gene *Pf4*, also known as *Cxcl4* (*Figure 5E*). *Pf4* was the first member of the chemokine family of small peptides to be discovered (*Deuel et al., 1977*). It is strongly expressed in platelet granules where it functions to promote blood coagulation, and is also expressed in monocyte/macrophage-lineage cells where it is involved in various physiologic and pathologic processes (*Abram et al., 2014*; *Calaminus et al., 2012*). In single cell and bulk transcriptomic studies, *Pf4* is highly enriched in all BAM subtypes (*Van Hove et al., 2019*) and is expressed early in EMP lineage commitment (*Matcovitch-Natan et al., 2016*; *Utz et al., 2020*). Our ribosomal profiling data suggested that *Pf4* would be a prime candidate gene for BAM-specific genetic labeling. To explore this, we analyzed the lineage of cells marked by *Pf4* expression in *Pf4-Cre* transgenic mice (*Pertuy et al., 2015*; *Tiedt et al., 2007*). In sharp contrast to *P2ry12CreER; Rosa26^{Ai14}* recombination, we found that in *Pf4-Cre; Rosa26^{Ai14}* mice, all CD206+LYVE1+ perivascular, pial and dural macrophages were recombined (*Figure 6A–B, D–H*). In the choroid plexus, we found a high degree of recombination that was specific to IBA1+ macrophages (80.9 ± 3.7% of IBA1+ cells) (*Figure 6C,I*). *Pf4-Cre; Rosa26^{Ai14}* mice also had a low degree of microglial recombination (5.24 ± 0.8% by flow cytometry, see *Figure 6F*), which appeared as clumps of cells scattered throughout the brain (asterisk in *Figure 6A*), suggestive of sporadic recombination and subsequent clonal expansion. We also saw consistent recombination in a small population of what appear to be neurons in the anterior amygdala (*Figure 6—figure supplement 1*). As has been reported, we found strong recombination in other organs, (not shown, *Pertuy et al., 2015*) and in platelets (*Figure 4—figure supplement 1E*). Altogether, we used a comparative ribosomal profiling approach to identify *Pf4* as a highly specific marker of BAMs (*Figure 5E*); based on this, we characterized the *Pf4-Cre* recombinase line and demonstrate that it is a robust method for genetic recombination in BAMs.

## Use of P2ry12-CreER in disease models

Some 'homeostatic' microglia signature genes, such as *P2ry12*, are downregulated with injury/disease or with aging (*Dubbelaar et al., 2018*; *van Wageningen et al., 2019*; *Wolf et al., 2017*). This raised the possibility that by inducing *P2ry12-CreER; Rosa26^{Ai14}* mice with tamoxifen prior to injury, microglia in these mice could be tracked by virtue of their expression of TdTomato, even though microglial P2RY12 expression may be reduced in areas of tissue damage. Furthermore, due to the specificity of *P2ry12-CreER*, we suspected that in the context of injury, *P2ry12-CreER* recombination could be used to distinguish homeostatic and disease-associated microglia from infiltrating myelogenous cells. To test this, we induced recombination in *P2ry12-CreER; Rosa26^{Ai14}* mice and then studied these mice in two separate injury models: unilateral MCAO (middle cerebral artery occlusion), a model of ischemic stroke, and EAE (experimental autoimmune encephalomyelitis), a model of neuroinflammatory injury similar to human multiple sclerosis.

For MCAO and EAE experiments, mice were induced with tamoxifen a week prior to injury, thus ensuring that despite any transcriptional changes in microglia post-injury, microglia would be genetically labeled and tracked by their TdTomato expression. Twenty-four hours after MCAO, mice were perfused, and brains were sectioned and immunostained for markers of microglial homeostasis (P2RY12, TMEM119). We also stained for CXCL10, a pro-inflammatory cytokine (*Giladi et al., 2020*;

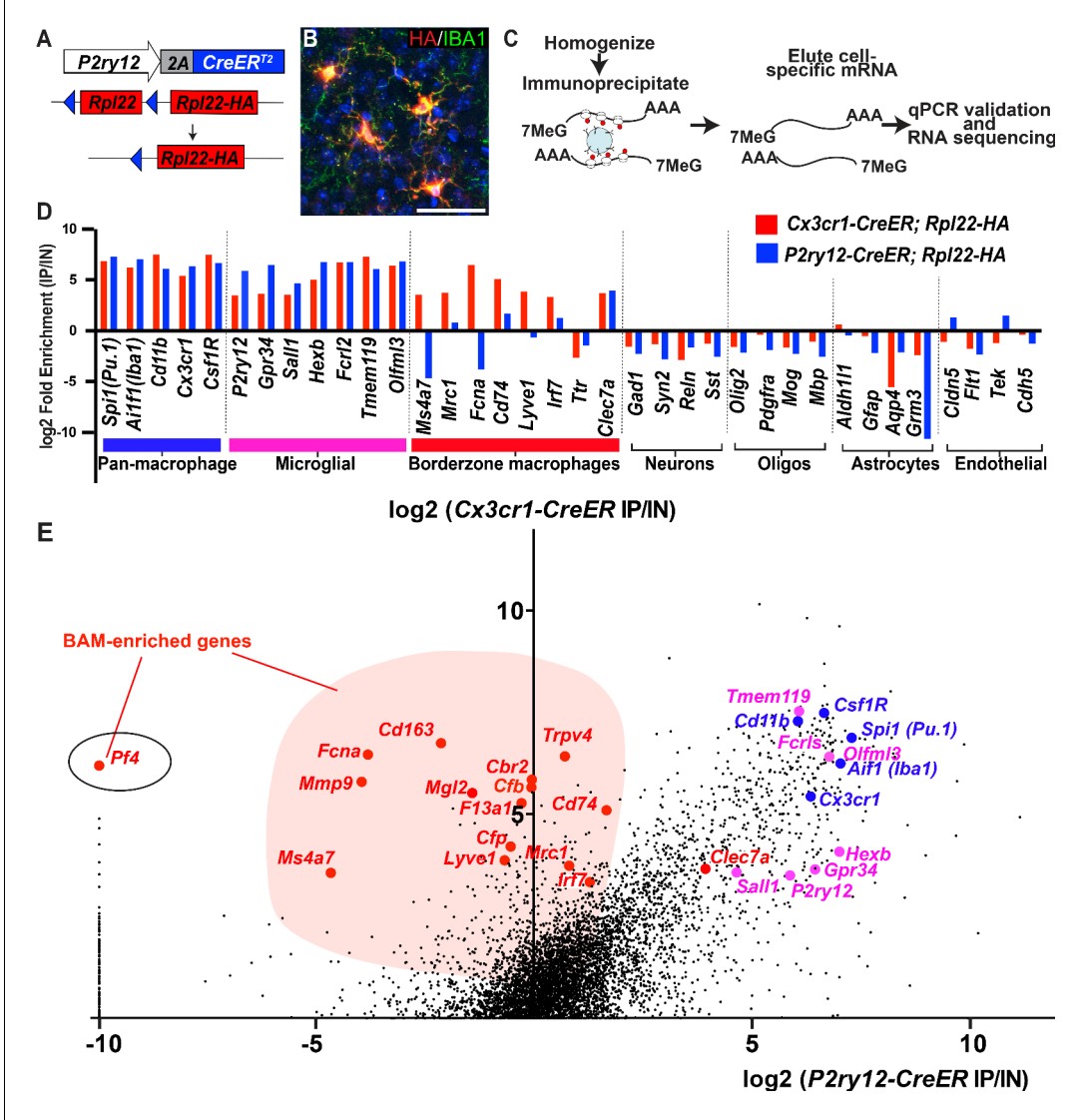

**Figure 5.** Cre-dependent ribosomal profiling of microglia. (**A**) Diagram illustrating *P2ry12-CreER*-dependent recombination of the *Rpl22-HA* allele, used to perform cell-type specific ribosomal profiling. (**B**) Immunostaining of tagged microglia for HA (red) and IBA1 (green) shows strong Cre-dependent expression of *Rpl22-HA* in microglia. (**C**) Workflow diagram for Cre-dependent ribosomal profiling experiments. (**D**) A comparison of relative enrichment for cell-type specific markers from *P2ry12-CreER; Rpl22-HA* and *Cx3cr1-CreER; Rpl22-HA* ribosomal immunoprecipitations. (**E**) Genome-wide comparisons of relative enrichment following immunoprecipitation of ribosomes in *P2ry12-CreER; Rpl22-HA* and *Cx3cr1-CreER; Rpl22-HA* mice. Pan-macrophage markers in blue; presumptive microglial markers in pink; presumptive BAM markers in red.

*Jordão et al., 2019*; *Wang et al., 2000*). Remarkably, ischemia-associated microglia (amoeboid-like CXCL10+ cells lacking P2RY12 and TMEM119 protein expression) filled the lesion core (*Figure 7A–E*). In the ischemic penumbra, outside of the stroke core, microglia had reduced process ramification, reduced P2RY12 and TMEM119 protein expression, and less frequent or reduced CXCL10 expression (*Figure 7C–E*).

To track microglia in the context of EAE, we used *P2ry12-CreER; Rosa26^{Ai14}; Cx3Cr1-EGFP* mice, and collected tissue samples 10 days post-EAE symptom onset. Spinal cords were collected, sectioned, and immunostained for EGFP and PU.1, a pan-myeloid transcription factor. At a gross level, we observed TdTomato+ microglia dispersed throughout inflammatory lesions (asterisk in *Figure 8A*). We observed a large number of amoeboid TdTomato+; Cx3cr1-EGFP+; PU.1+ microglia in EAE lesions (arrowhead in *Figure 8D*) in contrast to non-lesion areas of the spinal cord (*Figure 8C,E*). We also observed a large number of TdTomato-; PU.1+; Cx3cr1-EGFP- infiltrating

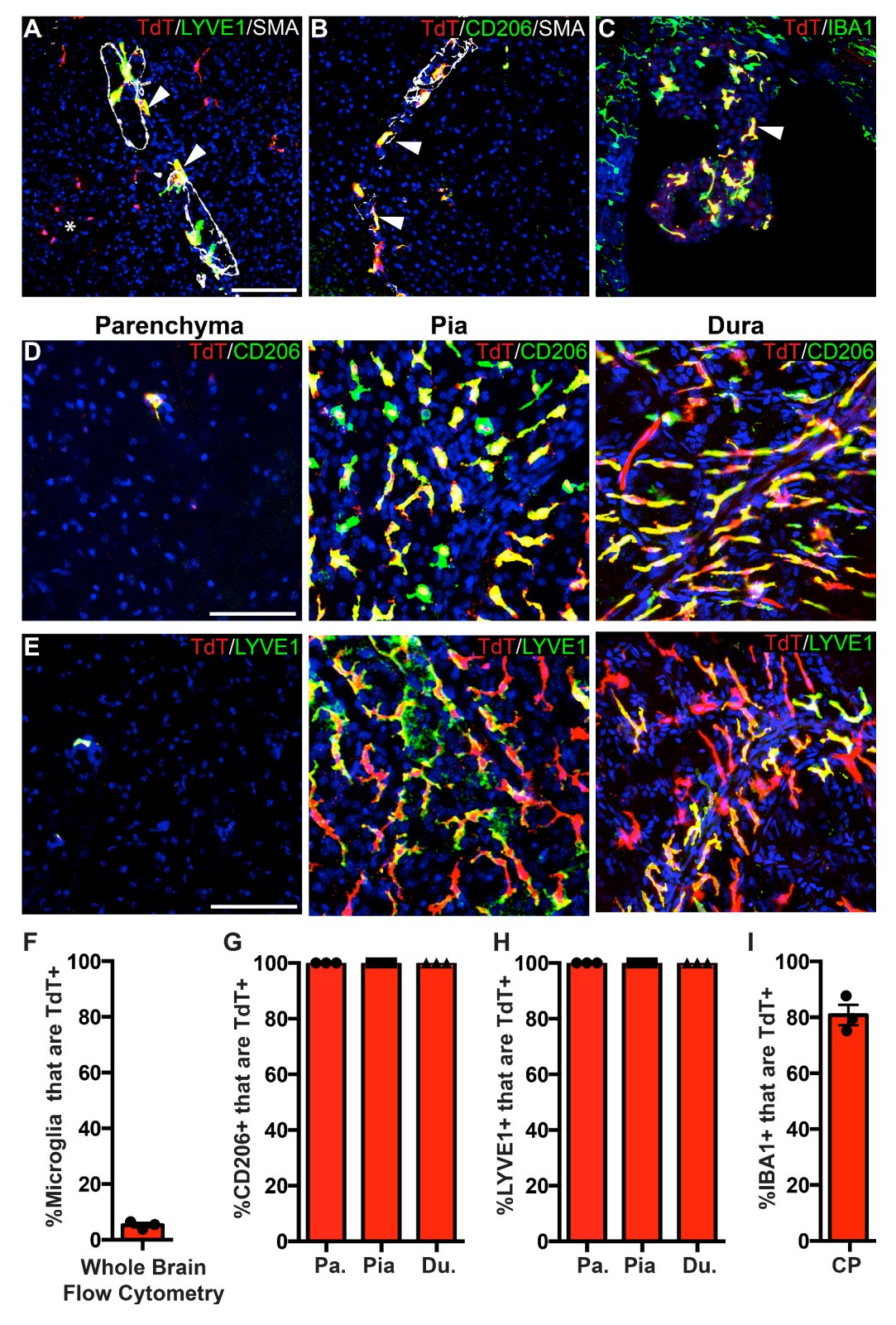

**Figure 6.** *PF4-Cre* robustly labels border-associated macrophages of the brain. (**A-C**) Analysis of *PF4-Cre; Rosa26*$^{Ai14}$ recombination in (**A**) LYVE1+ perivascular macrophages, (**B**) CD206+ perivascular macrophages, and (**C**) IBA1+ choroid plexus macrophages. Asterisk in **A**: recombination in a small cluster of microglia. (**D–E**) *PF4-Cre; Rosa26*$^{Ai14}$ whole-mount immunostaining of the cerebral cortex parenchyma, pia and dura for CD206 (**D**) and LYVE1 (**E**). (**F**) Flow cytometry quantification of microglial recombination in *PF4-Cre; Rosa26*$^{Ai14}$ mice. (**G–H**) Quantification of *PF4-Cre; Rosa26*$^{Ai14}$

*Figure 6 continued on next page*

*Figure 6 continued*

recombination in (G) CD206+ and (H) LYVE1+ perivascular, pial and dural macrophages. (I) Quantification of *PF4-Cre; Rosa26^{Ai14}* recombination in IBA1+ cells of the choroid plexus. Error bars = SE (F–H), SEM (I). Scale bars = A-E-100um.

The online version of this article includes the following source data and figure supplement(s) for figure 6:

**Source data 1.** Pf4-Cre robustly labels border-associated macrophages of the brain.
**Figure supplement 1.** *PF4-Cre* recombination in non-myeloid brain parenchymal cells.

myelogenous cells (asterisk in *Figure 8D*) and a smaller number of TdTomato-; PU.1+; Cx3cr1-EGFP+ potential infiltrating monocytes (open arrowhead in *Figure 8D*).

## Discussion

### Efficiency and specificity of P2ry12-CreER

Here, we describe a new method for the genetic manipulation of microglia in the CNS. We show that *P2ry12-CreER* robustly and specifically recombines microglia using flow cytometry, immunohistochemistry and ribosomal profiling. We did, however, find recombination in a subset of dural and choroid plexus macrophages. We believe that these recombined cells in the choroid plexus are likely to be Kolmer's epiplexus cells, which express significant levels of *P2ry12* (*Van Hove et al., 2019*). Whether the gene expression similarities between CNS microglia and Kolmer's epiplexus cells have any developmental basis or functional significance will be an interesting topic for future research. In the blood, we did not see any significant recombination of monocytes or platelets. In most organs of adult mice, there was little to no recombination, although in the spleen we did see significant levels of recombination in marginal zone macrophages. In embryos, we found that *P2ry12-CreER* robustly labels microglia, as well as a small fraction of meningeal and perivascular macrophages. We did not see recombination in any non-macrophage cell types in the embryonic brain. In addition, we did not observe recombination in the embryonic heart and liver, and found it only in a small fraction of lung and intestinal macrophages. Altogether, our data suggest that *P2ry12-CreER* is robust in embryonic and adult microglia, while recombination in non-CNS organs and border-associated macrophages is limited or not detected.

Of note, while we observed limited *P2ry12-CreER* recombination in CD206+ and LYVE1+ macrophages of the embryonic meninges, we saw significantly more recombination in CD206+ macrophages of the hippocampus. Interestingly, CD206+; TdTomato+ cells were most highly concentrated at the junction between the hippocampus and the overlying meninges. Continuous with the hippocampus, CD206+; TdTomato+ cells formed a 'stream' through the cortical SVZ, suggestive of preferential localization, or possibly migration. Possibilities for what these cells represent could fit a few different models: 1) 'transitionary' macrophages that express *P2ry12* while in border areas, which downregulate BAM genes as they enter the brain parenchyma; 2) developmental expression (or possibly continual expression, in the case of *P2ry12-CreER; Rosa26^{Ai14}* recombined dural macrophages) of *P2ry12* in a subset of BAMs; or 3) expression of *P2ry12* in cells that are transitioning from microglial to BAM identity. The turnover and exchange of BAMs and microglia into and out of their specific niches, and the gene expression programs that these cells adopt, could in theory be quite dynamic. Related to these observations, a recent study found that CD206+ and CD206- macrophages of the CNS derive from distinct yolk sac progenitors (*Utz et al., 2020*). Further lineage studies, ideally with BAM-specific *CreER* lines such as *Cd206-CreER* (*Nawaz et al., 2017*) (See *Supplementary file 3* for a list of BAM-targeting mouse lines) should provide further light on this matter.

### Comparison of P2ry12-CreER to existing microglial recombinase lines

In recent years, knock-in and transgenic mouse lines using the fractalkine receptor gene *Cx3cr1* (e.g. *Cx3cr1-Cre*, *-CreER* and *-eGFP*) have been some of the best and most frequently used tools for targeting microglia (*Goldmann et al., 2013*; *Parkhurst et al., 2013*). However, *Cx3cr1*-based mouse lines mark BAM macrophages in addition to microglia. *Cx3cr1* is also expressed in circulating T cells, B cells, NK cells and monocytes, which means that in the context of injury, *Cx3CR1-Cre, CreER* and *EGFP* are incapable of distinguishing between resident macrophages and other invading immune

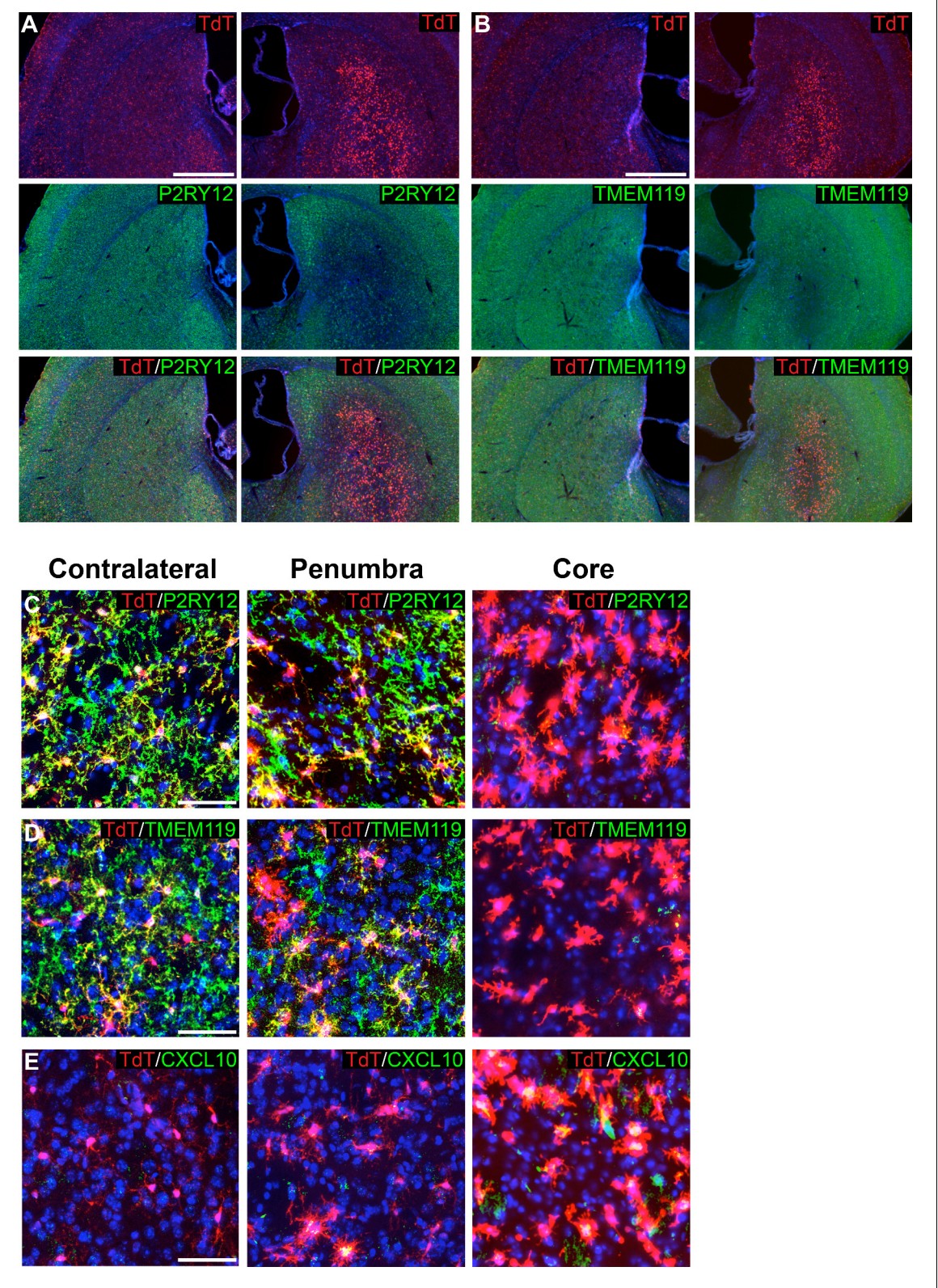

**Figure 7.** Genetic labeling of microglia during MCAO-induced stroke. (A-B) Genetically labeled microglia analyzed following MCAO-induced stroke show significant downregulation of microglial markers P2RY12 (A) and TMEM119 (B) in the ischemic core and penumbra. For both (A) and (B), the contralateral control hemisphere is the left column, the stroke-affected ipsilateral hemisphere is the right column. (C–D) Higher magnification of contralateral control area of the striatum, the stroke penumbra and core. P2RY12 and TMEM119 expression is reduced in the penumbra, and almost

*Figure 7 continued on next page*

*Figure 7 continued*

completely lost in the stroke core. In the stroke core, microglia are larger and amoeboid. (**E**) The cytokine CXCL10 is significantly upregulated in microglia of the stroke core. Scale bars = 500 μm (**A,B**), 50 μm (**C–E**).

cells (*Bar-On et al., 2010*; *Fogg et al., 2006*; *Imai et al., 1997*; *Liu et al., 2009*). The tamoxifen-inducible *Cx3cr1-CreER* allele avoids this problem to a degree, but requires a significant waiting period after recombination for recombined circulating cells to be depleted. This waiting period is inconvenient for most models of CNS injury and is incompatible with time-sensitive studies, such as models of neonatal brain injury. In addition to issues regarding specificity, *Cx3cr1* knock-in alleles (*EGFP, Cre, and CreER*) (*Goldmann et al., 2013*; *Parkhurst et al., 2013*) are also haploinsufficient for Cx3CR1, as they disrupt the endogenous *Cx3cr1* gene. The effects of the reduction of Cx3cr1 dosage may in turn affect microglial function (*Rogers et al., 2011*; *Hickman et al., 2019*; *Lee et al., 2010*).

Aside from *Cx3cr1*, knock-in *EGFP* and *CreER* lines have been generated for the microglial-enriched transcription factor *Sall1* (*Buttgereit et al., 2016*). However, caveats for these lines exist, including the fact that *Sall1-EGFP* and *CreER* both disrupt the function of the *Sall1* gene, loss of which disrupts microglial identity, see *Buttgereit et al., 2016*. *Sall1-CreER* also marks a broad range of non-myeloid CNS cell types (*Chappell-Maor et al., 2019*; *Stifter and Greter, 2020*), possibly due to tamoxifen-independent recombination and strong neuroectodermal expression of *Sall1* during development (*Harrison et al., 2012*).

In addition to *Cx3CR1* and *Sall1*-based microglial recombinase lines, new *Tmem119*-targeted mouse alleles have been generated (*Tmem119-GFP, RFP, CreER*) (*Kaiser and Feng, 2019*; *Ruan et al., 2020*). In terms of specificity, both *Tmem119-CreER* and *–GFP* appear to show low but not insignificant levels of recombination or expression in brain-derived CD11b+; C45int/hi myeloid cells (1.9% and 6.9% for *CreER* recombination and GFP expression respectively). In the circulating blood, no *Tmem119-GFP*(+) monocytes were reported, but 7 days after the last tamoxifen injection, 3% of circulating CD45(+) monocytes were labeled by *Tmem119-CreER*. Of note, this waiting period of 7 days differed from our blood flow cytometry analysis, which was performed the day after the last tamoxifen dose. In the neonatal P1 brain, *Tmem119-GFP* showed widespread expression over-lapping with the vascular endothelial cell marker, CD31. In adulthood, prolonged administration of tamoxifen to *Tmem119-CreER* mice induced recombination in non-IBA1(+) cells. This recombination was quite strong in IBA1(-) meningeal cells and in cells adjacent to large penetrating blood vessels, likely representing meningeal and perivascular fibroblasts, respectively. Indeed, based on single cell sequencing, *Tmem119* is highly expressed in CNS resident fibroblasts (http://betsholtzlab.org/Vas-cularSingleCells/database.html).

In contrast to these other models, the *P2ry12-CreER* line has a number of advantages: 1) the design of the *P2ry12-CreER* allele results in maintained gene expression from the endogenous locus, avoiding haploinsufficiency or null allele complications; 2) recombination is absent in non-myeloge-nous cells in the adult CNS; 3) recombination is absent in the adult circulation; 4) recombination in the embryonic mouse brain is robust, with low levels of recombination in CD206+ and LYVE1+ cells; 5) *P2ry12-CreER* has very low levels of background recombination; and 6) recombination in non-CNS organs in the adult and embryo appears to be limited, except for moderate levels of recombination observed in marginal zone macrophages of the adult spleen. In future studies, it will be interesting to see if additional microglial markers, such as *Gpr34*, *P2ry6*, *P2ry13*, *HexB*, *Siglec-H*, prove useful for recombinase-based microglial targeting (*Hickman et al., 2013*; *Konishi et al., 2017*).

## P2ry12-CreER applications

In addition to gene deletion studies or reporter-based tracking of microglia, one particularly useful application of the *P2ry12-CreER* line is for Cre-dependent ribosomal profiling of microglia. Here, using a subtractive approach, we found a number of genes that had very high enrichment in *Cx3cr1-CreER*; *Rpl22-HA* samples but no enrichment in *P2ry12-CreER*; *Rpl22-HA* samples. Interestingly, some of these markers (*Mrc1*, *F13a1*, *Ms4a7* and *Mmp9*) were recently identified in other studies that used physical dissociation of the dural membranes from the neural parenchyma to separate BAMs from other CNS macrophage subtypes (*Jordão et al., 2019*; *Van Hove et al., 2019*). One

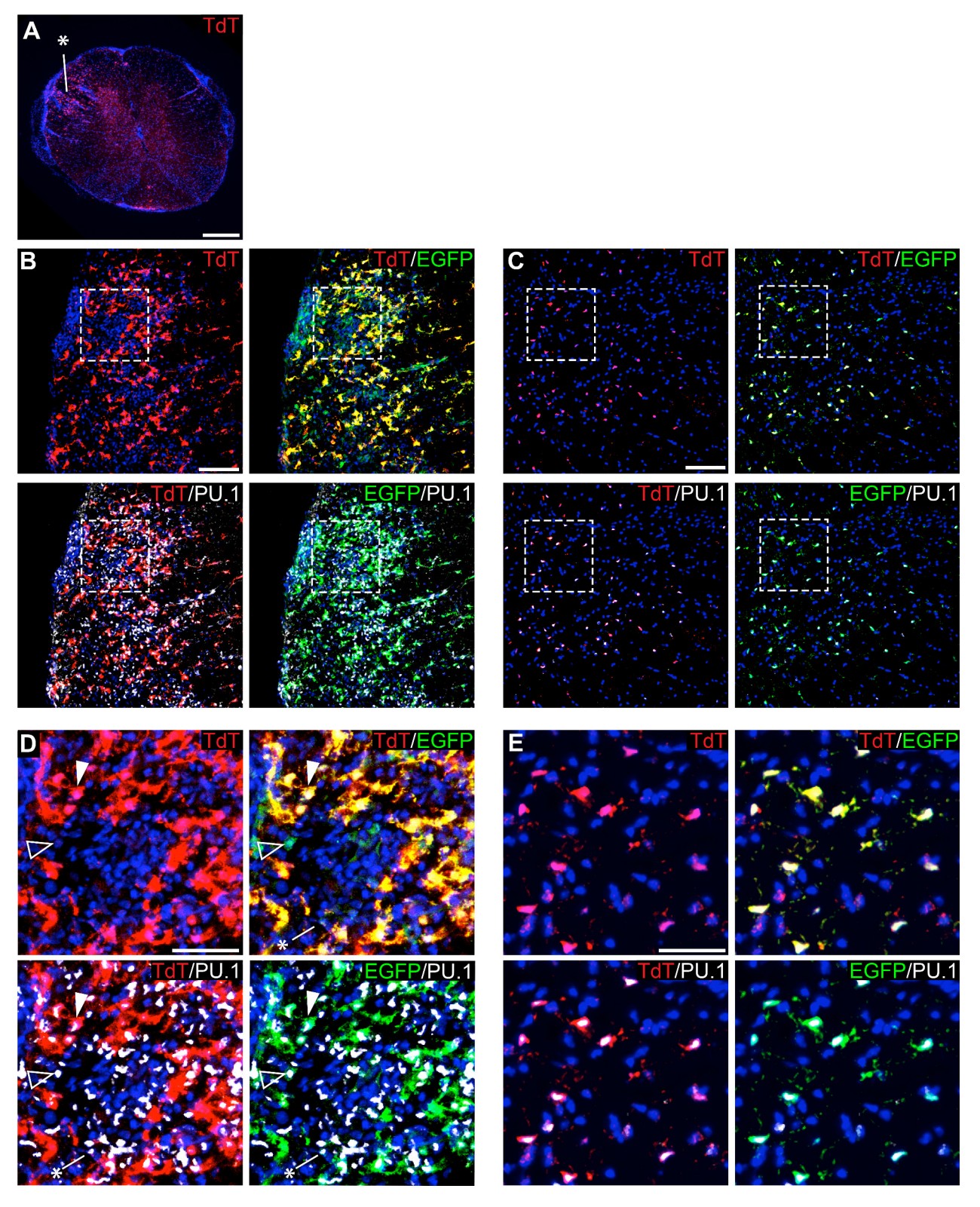

**Figure 8.** Genetic labeling of microglia during EAE. (**A**) Spinal cord microglia labeled prior to induction of EAE are dispersed throughout inflammatory lesions (see asterisk). (**B**) EAE spinal cord lesion in a *P2ry12-CreER; Rosa26^Ai14; Cx3cr1-EGFP* mouse that was given tamoxifen a week prior to EAE induction. Sections were immunostained for PU.1 (white) and Cx3cr1-EGFP (green). (**C**) Control, non-lesioned area of the spinal cord. (**D**) Higher magnification view of EAE lesion, showing microglia (closed arrowhead), Cx3cr1-EGFP; TdTomato(-) potential infiltrating monocytes (open arrowhead),
*Figure 8 continued on next page*

*Figure 8 continued*

and TdTomato(-); EGFP(-); PU.1 (+) infiltrating myelogenous cells (asterisk). (**E**) Higher magnification view of a non-lesioned area of the spinal cord. Scale bars = 500 µm (**A**), 100 µm (**B,C**), 40 µm (**D,E**).

major drawback to strategies that rely on the physical dissection of these populations is that this process results in significant levels of cross contamination from microglia and other cells that that are in close contact with the meningeal layers. Furthermore, physical dissection of the meningeal layers does not separate perivascular macrophages of the penetrating vessels from the parenchymal microglia. In contrast, *P2ry12-CreER* recombination provides a more precise and complete genetic segregation of these populations that minimizes the transcriptional changes that are known to occur during sorting. The differential enrichment observed in our subtractive approach revealed several transcripts only expressed in BAMs. We focused on one such gene, *Pf4*, and found that *Pf4-Cre* robustly and preferentially recombines these cells.

The *P2ry12-CreER* mouse line is also particularly well suited for the labeling and tracking of microglia during CNS injury. One major barrier to studying microglia in the context of injury is that microglia undergo significant transcriptional changes in response to injury and/or inflammation, making it difficult to distinguish these microglia from other macrophage subtypes (*Haimon et al., 2018*; *Jordão et al., 2019*). These transcriptional changes are also induced by methods used to prepare microglia for analysis by flow cytometry. By genetically labeling and profiling microglia with *P2ry12-CreER* and TRAP-seq, these transcriptional changes associated with flow cytometry can be avoided (*Haimon et al., 2018*). Using *P2ry12-CreER*, transcriptional profiling or histochemical analysis of microglia can be performed at different stages of CNS injury, from the initial inflammatory response to later regenerative stages. Furthermore, due to the specificity of *P2ry12-CreER*, the contributions of CNS microglia can be parsed separately from invading peripheral monocytes.

In contrast to our EAE and MCAO microglial labeling strategy, in which microglia were labeled a week prior to injury, microglia could also be targeted in a tamoxifen-dependent manner during neuroinflammation. Although we have not yet tested this, we suspect that *P2ry12-CreER* recombination is likely to be reduced in inflammation-associated microglia, due to downregulation of *P2ry12*. Tamoxifen-dependent effects on neuroinflammation will be important to control for if this strategy is used (*Bebo et al., 2009*).

## Conclusions and further directions

Here, we show by immunohistochemistry, flow cytometry, and ribosomal profiling that *P2ry12-CreER* has a number of advantages over existing microglial targeting *Cre* and *CreER* mouse lines. Using data derived from our Cre-dependent ribosomal profiles of microglia and analysis of the *Pf4-Cre* mouse line, we also show that *Pf4* is a highly specific marker of border-associated macrophages. We believe that the *P2ry12-CreER* allele will be particularly useful for conditional gene ablation in microglia, Cre-dependent transcriptional profiling, or reporter tracking studies of microglia. Going forward, new recombinase lines and new intersectional targeting strategies, such as dual Cre+Flp-dependent reporters (*Dymecki et al., 2010*), will be critical for parsing the functional role of specific myeloid subpopulations in development and disease.

## Materials and methods

**Key resources table**

| Reagent type (species) or resource | Designation | Source or reference | Identifiers | Additional information |
|---|---|---|---|---|
| Strain, strain background *Mus musculus* | *Cx3cr1-CreER* | JAX | Stock No: 020940. MGI: 5467985. | |
| Strain, strain background *Mus musculus* | *Pf4-Cre* | JAX | Stock No: 008535. MGI: 3764698. | |

*Continued on next page*

*Continued*

| Reagent type (species) or resource | Designation | Source or reference | Identifiers | Additional information |
|---|---|---|---|---|
| Antibody-linked beads | Protein A Dynabeads | Invitrogen | Catalog No: 10001D. | |
| Chemical | Cyclohexamide | Sigma | Catalog No: C7698-1G. | |
| Molecular biology reagent | RNAse Inhibitor | New England BioLabs | Catalog No: M0314S. | |
| Molecular biology reagent | Protease Inhibitors Cocktail | Sigma | Catalog No: S8830. | |
| Antibody | Anti-CD11b-PECy7 | BD Biosciences | Rat monoclonal antibody. Catalog No: 561098. RRID:AB_2033994. | Used at 1:100. |
| Antibody | Anti-CD206 | Biorad | Rat monoclonal antibody. Catalog No: MCA2235T RRID:AB_1101333. | Used at 1:150. |
| Antibody | Anti-CD45-APC | BD Biosciences | Rat monoclonal antibody. Catalog No: 559864. RRID:AB_398672. | Used at 1:100. |
| Antibody | Anti-CXCL10 | R and D Systems | Goat polyclonal antibody. Catalog No: AF-466-NA. RRID:AB_2292487. | Used at 1:300. |
| Antibody | Anti-EGFP | Abcam | Chicken polyclonal antibody. Catalog No: ab13970. RRID:AB_300798. | Used at 1:300. |
| Antibody | Anti-ERTR7 | Abcam | Rat monoclonal antibody. Catalog No: Ab51824. RRID:AB_881651. | Used at 1:150. |
| Antibody | Anti-HA | Cell Signaling | Rabbit monoclonal antibody. Catalog No: C29F4. RRID:AB_1549585. | Used at 1:300. |
| Antibody | Anti-IBA1 | Novus | Goat polyclonal antibody. Catalog No: NB100-1028. RRID:AB_521594. | Used at 1:300. |
| Antibody | Anti-LYVE1 | Abcam | Rabbit polyclonal antibody. Catalog No: ab14917. RRID:AB_301509. | Used at 1:300. |
| Antibody | Anti-NEUN | Millipore | Mouse monoclonal antibody. Catalog No: MAB377. RRID:AB_2298772. | Used at 1:100. |
| Antibody | Anti-OLIG2 | Millipore | Rabbit polyclonal antibody. Catalog No: AB9610. RRID:AB_570666. | Used at 1:300. |
| Antibody | Anti-P2RY12 | A generous gift from Dr. David Julius. | Rabbit polyclonal antibody. | Used at 1:1000. |
| Antibody | Anti-PU.1 | Cell Signaling | Rabbit polyclonal antibody. Catalog No: 2266S. RRID:AB_10692379. | Used at 1:300. |

*Continued*

| Reagent type (species) or resource | Designation | Source or reference | Identifiers | Additional information |
|---|---|---|---|---|
| Antibody | Anti-SMA-647 | Santa Cruz Biotechnologies | Mouse monoclonal antibody. Catalog No: Sc-32251. RRID:AB_262054. | Used at 1:150. |
| Antibody | Anti-SOX9 | R and D Systems | Goat polyclonal antibody. Catalog No: AF3075. RRID:AB_2194160. | Used at 1:300. |
| Antibody | Anti-TdTomato | Chromotek | Rat monoclonal antibody. Catalog No:5F8. RRID:AB_2336064. | Used at 1:500. |
| Antibody | Anti-TdTomato | Promega | Rabbit polyclonal antibody. Catalog No: #632496. RRID:AB_10013483. | Used at 1:500. |
| Antibody | Anti-TMEM119 | Abcam | Rabbit monoclonal antibody. Catalog No: ab209064. RRID:AB_2800343. | Used at 1:100. |

## Mice

*P2ry12-CreER* mice were generated by CRISPR-facilitated homologous recombination. The *P2ry12* coding sequence was joined to *CreER* by deletion of the *P2ry12* stop codon and in-frame 3' insertion of *P2A-CreER*. The P2A fusion peptide contains a 5' GSG motif, which has been shown to significantly increase fusion protein cleavage efficiency (*Szymczak-Workman et al., 2012*). *P2ry12-CreER* mice were generated by pronuclear injection of the *P2ry12-CreER* targeting construct and a Cas9 +gRNA expression plasmid (Biocytogen) into fertilized mouse eggs, followed by adoptive embryo transfer. Founder mice and progeny were screened by PCR, and proper genomic integration was verified by Southern blot and genomic sequencing. Recombination was induced by three doses of tamoxifen dissolved in corn oil, administered by oral gavage every other day (150 µL of 20 mg/mL). For embryonic mouse inductions, pregnant dams were given tamoxifen (150 µL of 20 mg/mL) on E13.5, E15.5 and E17.5, for a total of three gavage injections. All mouse work was performed in accordance with UCSF Institutional Animal Care and Use Committee protocols. Mice had food and water ad libitum. The *P2ry12-CreER* mouse line will be deposited at Jackson Labs (Stock #034727) and at the Mutant Mouse Resource and Research Center (MMMRC).

## Histology and immunostaining

Mouse brains and other organs were harvested following transcardial perfusion with 20 mL cold PBS and 20 mL cold 4% formaldehyde. Tissue was fixed overnight at 4 degrees in 4% formaldehyde, followed by overnight incubation in 30% sucrose. Samples were embedded (Tissue Plus O.C.T. Compound, Fisher Scientific) and sectioned at 40 µm for adult tissues, and 20 µm for embryonic tissues. Whole-mount thin cortical sections and skull preparations were not incubated in sucrose, and immunostaining was started after a 2 hr 4% PFA post-fixation following perfusion and dissection. Sections or wholemount samples were immunostained using a blocking/permeabilization buffer of PBS containing 2% BSA, 5% donkey serum and. 5% TritonX-100. Primary and secondary antibodies were diluted in PBS containing 1% BSA and. 25% TritonX-100. Secondary antibodies conjugated to Alexa fluorophores were used at 1:300 (Jackson ImmunoResearch). Primary antibodies used included: SMA-647 (mouse monoclonal sc-32251, Santa Cruz Biotechnology, used at 1:150); Cd11b-PECy7 (BD Biosciences, used at 1:100); Cd45-APC (BD Biosciences, used at 1:100); CD206 (rat monoclonal, BioRad MCA2235T, used at 1:150); CXCL10 (goat polyclonal, R and D Systems AF-466-SP, used at 1:300); (ERTR7 (rat monoclonal, abcam ab51824, used at 1:300); EGFP (chicken polyclonal, abcam ab13970 used at 1:300); TdTomato (rat monoclonal, Chromotek, 5F8, used at 1:500); TdTomato (rabbit polyclonal, Living Colors DsRed Polyclonal #632496, used at 1:1000); P2ry12 (rabbit

polyclonal, kindly provided by Dr. David Julius, used at 1:1000); Iba1 (goat polyclonal, Novus NB100-1028, used at 1:300); Olig2 (rabbit polyclonal, Millipore, used at 1:300); NeuN (mouse monoclonal, Millipore MAB377, used at 1:100); Sox9 (goat polyclonal, R and D System AF3075, used at 1:300); Lyve1 (rabbit polyclonal, Abcam ab14917, used at 1:300); Pu.1 (rabbit polyclonal, Cell Signaling 2266S, used at 1:300); Tmem119 (rabbit monoclonal, Abcam ab209064); HA (rabbit monoclonal, Cell Signaling C29F4, used at 1:300). Cell counting in *Figure 1D* was performed on three mice, using three separate images taken from the spinal cord, cerebellum, cortex and hippocampus of each mouse.

## Flow cytometry

Microglial flow cytometry was performed using a percol gradient isolation strategy. To isolate single cells, brains were cut into small pieces and passed through a 40 µm filter. Single cell suspensions were prepared and centrifuged over a 30%/70% discontinuous Percoll gradient (GE Healthcare) and mononuclear cells were isolated from the interface. Flow cytometry was performed on a FACS Aria III using the FACSDiva 8.0 software (BD Biosciences) and was analyzed using FlowJo v10.6.1. TdTomato+ cells were analyzed for CD45 and CD11b expression following initial gatings for forward scatter, singlets and live cells. Blood smears and flow cytometry were performed on adult mice following three tamoxifen inductions as described above. Blood was collected from the facial vein and smears were imaged for EGFP and TdTomato expression. Prior to immunostaining, blood samples used for flow cytometry were depleted of erythrocytes using an ammonium chloride lysis buffer (.15M NH4Cl, 10 mM NaHCO3, 1.2 mM EDTA) wash. Blood flow cytometry was performed on a BD FACSVerse machine and analyzed using FlowJo v10.6.1.

## Ribosomal immunoprecipitations

Whole brains of *P2ry12-CreER; Rpl22-HA* and *Cx3cr1-Cre; Rpl22-HA* mice were dounce homogenized in 5 mL of homogenization buffer (1% NP-40, 0.1M KCL, 50 mM Tris pH 7.4, 0.02 mg/mL cycloheximide, 0.012M MgCl2, RNAse inhibitors, Heparin, 0.5 mM DTT). Tissue lysates were centrifuged at 10k-rcf for 10 min at four degrees to obtain a supernatant ribosomal fraction. An 80 µL input sample was taken at this point to serve as a comparison to immunoprecipitated samples. Rabbit monoclonal anti-HA antibody (Cell Signaling C29F4, 2.5 µL per mL) was added to the supernatant and incubated for 4 hr at four degrees. Iron-linked Protein A beads were added to the ribosomal fraction/antibody mixture (40 µL per mL) and incubated for an additional 4 hr before being magnetically precipitated. Immunoprecipitate/beads were washed with a high salt buffer (0.3M KCL, 1% NP-40, 50 mM Tris pH 7.4, 0.012M MgCL2, cyclohexamide, 0.5 mM DTT) three times and then eluted with 350 µL of buffer RLT (Qiagen RNeasy kit). Input and immunoprecipitated RNA was purified on column (Qiagen RNeasy kit) and frozen as aliquots at −80 degrees. cDNA from ribosomal immunoprecipitations was generated using Maxima First Strand cDNA Synthesis Kit (Thermo scientific) and qPCRs were performed using SYBR Green qPCR Mastermix (Bimake) on a BioRad CFX Connect thermocycler. Quantitative comparisons between immunoprecipitations and input samples were made using delta-delta Ct normalization, with GAPDH as a normalizing control.

## Library preparation and sequencing

Total RNA quality was assessed by spectrophotometer (NanoDrop, Thermo Fisher Scientific Inc, Waltham, MA) and with an Agilent Fragment Analyzer (Agilent Technologies, Palo Alto, CA). All RNA samples had integrity scores greater than 9, reflecting high-quality, intact RNA. RNA sequencing libraries were generated using the Nugen Universal Plus mRNA-Seq kit with multiplexing primers, according to the manufacturer's protocol (NuGen Technologies, Inc, San Carlos, CA). Fragment size distributions were assessed using Agilent's Fragment Analyzer DNA high-sensitivity kit. Library concentrations were measured using KAPA Library Quantification Kits (Kapa Biosystems, Inc, Woburn, MA). Equal amounts of indexed libraries were pooled and single-end 50 bp sequencing were performed on the Illumina HiSeq 4000 at the UCSF Center for Advanced Technology (CAT) (http://cat.ucsf.edu).

### RNA sequencing analysis

Raw reads were pseudo-mapped to the mouse transcriptome using Salmon 0.13.1 (*Patro et al., 2017*). Quantified transcripts were passed to Tximport 1.12 for gene-level summarization and differential expression analysis was performed with DESeq2 1.24, using alpha = 0.05 (*Love et al., 2014*; *Love et al., 2015*). Genes with total read counts below 20 in all samples were eliminated from the analyses. Data have been submitted to the Gene Expression Omnibus (GEO) repository for datasets; the accession number for this dataset is: GSE138333.

### MCAO

MCAO was performed in spontaneously breathing animals anesthetized with 3% isofluorane in 100% $O_2$. The right internal carotid artery (ICA) was dissected and a temporary ligature was tied using a strand of 6–0 suture at its origin. This ligature was retracted laterally and posteriorly to prevent retrograde blood flow. A second suture strand was looped around the ICA above the pterygopalatine artery and an arteriotomy was made proximal to the isolated ICA. A silicone coated 6–0 nylon filament from Doccol Corporation (Sharon, MA, USA) was inserted 5 mm to occlude the MCA and the second suture strand was tied off to secure the filament for the duration of occlusion. Following recovery from anesthesia, mice were returned to their cage for the 90 min duration of the occlusion. Injury was confirmed by severe left frontal/hindlimb paresis resulting in circling movements during the occlusion period. For reperfusion, each animal was anesthetized and all suture ties and the occluding filament were removed. Avitene Microfibrillar Collagen Hemostat (Warwick, RI, USA) was placed over the arteriotomy and the skin incision was closed. Staining for P2RY12, TMEM119, and CXCL10 was performed as described in 'Immunohistochemistry'.

### EAE

*P2ry12-CreER; Rosa26^{Ai14};Cx3cr-1-EGFP* mice (C57BL/6 background) were subcutaneously injected with 200 µl of an emulsion of Complete Freund's adjuvant, MOG35-55 peptide, and inactivated mycobacterium tuberculosis (EK-2110; Hooke Laboratories). 2 hr and 24 hr afterwards, 200 ng of Pertussis toxin was injected i.p. (EK-2110; Hooke Laboratories). At 10 d after onset of symptoms, mice were perfused (see Histology and immunostaining above). Following perfusion, spinal cords were dissected and incubated in 30% sucrose, overnight. The following day, spinal cords were frozen in OCT and 20 µm sections were generated. Staining for PU.1, and EGFP was performed as described in 'Immunohistochemistry'.

### Imaging

Confocal images were taken using a motorized Zeiss 780 upright laser scanning confocal microscope with a 34 detector array with a water immersion Zeiss Plan Apochromat 20X/1.0 D = 0.17 Parfocal length 75 mm (Zeiss, Germany). Images for *Figure 2* and *Figure 3C* were taken with a Zeiss Axio Imager.Z2 epifluorescent microscope.

### Statistical analysis

Sample size was not precalculated using a statistical framework, but conformed to general standards in the field. Sample analysis was not blinded. For all immunostaining quantification, values for each mouse were calculated by averaging three pictures from each mouse. For flow cytometry in *Figure 4D*, four separate mice were used per genotype. Differences between means were compared using a two-tailed t-test, with an alpha of 0.05. For statistical analysis in *Figure 1—figure supplement 1A, C*, a One-Way ANOVA with a Tukey's multiple comparisons test was performed, with an alpha of 0.05.

For ribosomal immunoprecipitations, three samples were prepared, each of which was derived from a separate single mouse brain preparation (three immunoprecipitations from three separate mouse brains). Input samples were created by collecting 80 µL samples of supernatant from each of these brain preparations, prior to immunoprecipitation. Samples from separate mice were considered biological replicates. We did not eliminate any samples as outliers from our analysis. Differences in gene expression were tested using DESeq2, with an alpha of 0.05, with sample pairing in the model.

## Acknowledgements

This study was supported in part by HDFCCC Laboratory for Cell Analysis Shared Resource Facility through a grant from NIH (P30CA082103). NS is supported by AHA Postdoctoral fellowship 20POST35120371. We would like to thank UCSF genomics core members Andrea Barczak, Matthew Aber, and Joshua Rudolph for their help in generating ribosomal immunoprecipitation libraries for our microglial transcriptional profiling experiments. We would also like to thank Erik Chow, Kaitlin Chaung and Derek Bogdanoff at the UCSF Center for Advanced Technology (http://cat.ucsf.edu) for their help with high-throughput sequencing. We would also like to thank Marie La Russa, Daniel Bayless and Joseph Knoedler for helpful manuscript editing suggestions.

## Additional information

### Funding

| Funder | Grant reference number | Author |
|---|---|---|
| National Institutes of Health | K08NS96192 | Thomas D Arnold |
| American Heart Association | 20POST35120371 | Nicolas Santander |

The funders had no role in study design, data collection and interpretation, or the decision to submit the work for publication.

### Author contributions

Gabriel L McKinsey, Conceptualization, Resources, Data curation, Formal analysis, Supervision, Validation, Investigation, Visualization, Methodology, Writing - original draft, Project administration, Writing - review and editing, Performed flow cytometry experiments; Carlos O Lizama, Investigation, Performed flow cytometry experiments; Amber E Keown-Lang, Investigation, Assisted with the sectioning of a subset of the adult organs; Abraham Niu, Investigation, Assisted with a subset of the mouse perfusions and brain sectioning for Figure 6; Nicolas Santander, Formal analysis, Investigation, Writing - review and editing; Amara Larpthaveesarp, Resources, Investigation, Methodology; Elin Chee, Investigation, Assisted with mouse husbandry during the initial stages of the project; Fernando F Gonzalez, Resources, Funding acquisition; Thomas D Arnold, Conceptualization, Resources, Supervision, Funding acquisition, Visualization, Methodology, Project administration, Writing - review and editing

### Author ORCIDs

Gabriel L McKinsey (iD) https://orcid.org/0000-0002-5503-2830
Nicolas Santander (iD) https://orcid.org/0000-0001-8919-833X

### Ethics

Animal experimentation: All mouse work was performed in accordance with UCSF Institutional Animal Care and Use Committee protocols (#AN177934-01).

### Decision letter and Author response

Decision letter https://doi.org/10.7554/eLife.54590.sa1
Author response https://doi.org/10.7554/eLife.54590.sa2

## Additional files

### Supplementary files

• Source code 1. DeSeq2 source code. Source code used for differential expression analysis. For details, please see *Love et al., 2014*, *Love et al., 2015*.

• Source code 2. Salmon pseudomapping source code. Source code used for mapping reads to mouse transcriptome. For details, please see *Patro et al., 2017*.

• Supplementary file 1. Microglial markers analyzed in Tabula Muris. A short list of microglial enriched genes that were analyzed in the Tabula Muris. Of these genes, *P2ry12* was selected for further analysis.

• Supplementary file 2. Results of comparative DESeq2 analysis of *P2ry12-CreER* and *Cx3cr1-CreER* ribosomal immunoprecipitations. For details please see 'RNA sequencing analysis' section in Materials and methods section.

• Supplementary file 3. CNS border-associated macrophage (BAM) recombinase lines. A short list of BAM-targeting recombinase and DTR lines.

• Transparent reporting form

## Data availability

Sequencing data have been submitted to the Gene Expression Omnibus (GEO) repository for datasets. The accession number for this dataset is: GSE138333.

The following dataset was generated:

| Author(s) | Year | Dataset title | Dataset URL | Database and Identifier |
|---|---|---|---|---|
| McKinsey G, Santander N, Arnold T | 2020 | Translational profiling of microglia | https://www.ncbi.nlm.nih.gov/geo/query/acc.cgi?acc=GSE138333 | NCBI Gene Expression Omnibus, GSE138333 |

The following previously published dataset was used:

| Author(s) | Year | Dataset title | Dataset URL | Database and Identifier |
|---|---|---|---|---|
| Haimon Z, Volaski A, Orthgiess J, Boura-Halfon S, Varol D, Shemer A, Yona S, Zuckerman B, David E, Chappell-Maor L, Bechmann I, Gericke M, Ulitsky I, Jung S | 2018 | Re-evaluating Microglia Expression Profiles Using RiboTag and Cell Isolation Strategies | https://www.ncbi.nlm.nih.gov/geo/query/acc.cgi?acc=GSE114001 | NCBI Gene Expression Omnibus, GSE114001 |

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
