## [Decision Letter]

**Acceptance summary:**

Microglia are the resident immune cells of the central nervous system that play critical roles in development, homeostasis and inflammation. This study reports the generation of *P2ry12-CreER* mice as a useful new tool to genetically target and study microglia biology. The authors used these mice to track microglia during development, following injury and inflammation.

**Decision letter after peer review:**

Thank you for submitting your article "New tools for genetically targeting myeloid populations in the central nervous system" for consideration by *eLife*. Your article has been reviewed by four peer reviewers, including Isaac M Chiu as the Reviewing Editor and Reviewer #1, and the evaluation has been overseen by Satyajit Rath as the Senior Editor. The following individuals involved in review of your submission have agreed to reveal their identity: Beth Stevens (Reviewer #3); Amanda Sierra (Reviewer #4).

The reviewers have discussed the reviews with one another and the Reviewing Editor has drafted this decision to help you prepare a revised submission.

Summary:

Microglia are important cells that regulate CNS homeostasis, immunity, and neurodegeneration. This manuscript provides an interesting new tool, the *P2RY12-CreER* mice to specifically label and target microglia and not other brain macrophages (meninges, choroid plexus, perivascular space). Once validated, this resource would be widely used in the field and the authors plan to make this new line available at Jackson Labs. All reviewers were enthusiastic about this new resource and mouse tool for the field. Although the data shown look promising, a more thorough validation and characterization of this new mouse line is needed. In particular, the authors must put a strong effort into actually quantifying the percentage of cells labeled/targeted in each experiment, instead of using terms such as "widespread or sparse labeling". There were also missing pieces of information on methodology used and references that should be addressed before publication. We list the concerns below.

Essential revisions:

1) Given the leakiness of some *CreER* strains in combination with certain reporter lines (see Van Hove et al., 2019 and Chappell-Maor et al., 2019), it is critical to analyze the recombination of microglia in *P2ry12-CreER Ai14* mice also without tamoxifen treatment.

2) More rigorous quantification and assessment of recombination in BAMs.

Perivascular macrophages can be interrogated in brain sections by IHC, but to achieve a complete assessment of the meningeal BAMs, the meninges need to be removed from inside of the skull. Only the pia layer of the meninges stays adhered to the brain when the skull is removed. LYVE1 is also not an appropriate marker for labeling of BAMs as it only labels a subset. Importantly, dounce homogenization followed by percoll purification is not an appropriate method for isolating BAMs and the expression of TdTomato in other cells than microglia can therefore not be quantified correctly using these approaches.

3) Effect on P2RY12 protein expression and function.

Figure 1A: The authors use single examples of immunostaining against P2RY12 to demonstrate that the *P2ry12-CreER* mouse has normal P2RY12 protein expression. However, this claim should be supported by quantification, and immunohistochemistry is not a sensitive method for measuring protein levels. It will be critical to validate that the P2RY12 protein expression in the *P2ry12-CreER* mice is similar to WT mice with and without tamoxifen treatment. This is straightforward as there are antibodies against P2RY12 commercially available for flow cytometry. Additionally, the P2RY12-dependent motility of microglial processes in the *P2ry12-CreER* mouse should be assessed. Over the last decade, evoking laser-induced injuries has become a standard paradigm in many labs that are studying microglia.

4) Effect on P2ry12 transcription.

Based on the (n=1) ribosomal immunoprecipitations, the *p2ry12* transcriptions might appear to be normal and based on this one example. It would be helpful if the authors could quantify P2ry12 transcript levels using qRT-PCR to confirm that the presence of the *P2ry12-CreER* allele does not impact expression of the gene.

5) Reproducibility of Ribosomal immunoprecipitations experiment.

This experiment is elegant and support the conclusion that *P2ry12-CreER* is selective to microglia. However, is this (n=1) experiment reproducible? Please repeat for a minimum of 3 replicate experiments. It would be very powerful to also perform ribosomal immunoprecipitation on the meninges.

6) In Figure 1B, CD11b and CD45 does not suffice to detect non-parenchymal macrophages by flow cytometry. CD11bhi cells also comprise monocytes, neutrophils, some cDCs and conversely, some borderzone macrophages are CD45low. More markers should be included to identify borderzone macrophages.

7) In Figure 1C, what is the percentage of microglia, borderzone macrophages and monocytes that are TdTomato+ in *P2ry12-CreERAi14* mice by flow cytometry?

8) In Figure 1E-G, the images suggest that *P2ry12-CreER;Ai14* recombination does not occur in perivascular or meningeal macrophages, nor in non-macrophage cells in the brain, but can be detected in choroid plexus macrophages. These conclusions should be supported by quantification, either of immunohistochemistry images, or of FACS data. In Figure 1G, what are the TdTomato+ cells in *P2ry12-CreER*; *Ai14* mice (Iba1-) in the choroid plexus?

9) In Figure 2, the authors describe recombination in *P2ry12-CreER*; *Ai14* embryos in microglia, in choroid plexus macrophages and also in some meningeal macrophages. Are perivascular macrophages also labeled? Also here, the images should be quantified. Are macrophages in other embryonic tissues also TdTomato+ or just brain macrophages? It seems that this *CreER* strain is not microglia-specific during embryogenesis. This should be discussed in more detail. They should discuss their model also in the context of other previously described models to target microglia (for example Kaiser et al., 2019).

10) Analyses and quantification of recombination in other organs.

In Figure 3, what is the percentage of the macrophages positive for TdTomato in the different organs? This should be quantified (FACS or immunohistochemistry). Where are they located (associated to vessels, peripheral nerves)? The authors state that recombination in tissue-resident macrophages in peripheral organs is “limited” even though they provide images of TdTomato expression in both the spleen, lung, heart, thymus, intestine, and liver. Some level of quantification should be provided to support that this expression is “limited” and negligible. The recombination in the brains of these mice (n = 3 or higher) should also be measured as a positive control to demonstrate high recombination efficiency.

11) In Figure 3A, there seems to be TdTomato expression in neurons. Can the authors comment on this or provide higher magnification images to exclude co-expression?

12) In Figure 3C, it is not clear why the cells are yellow as there is no GFP.

13) Some microglia signature genes including P2ry12 are downregulated upon microglia activation and in disease. Can this strain be utilized to target activated microglia?

14) P2ry12 is also highly expressed by platelets. Do the authors find evidence that megakaryocytes are targeted with the *P2ry12-CreER*;*Ai14* mice?

15) They show that with Pf4-Cre mice borderzone macrophages can be targeted. This needs better characterization and should be quantified. Are monocytes, tissue macrophages in other organs, or any other cells also labeled?

[Editors' note: further revisions were suggested prior to acceptance, as described below.]

Thank you for re-submitting your article "A New Genetic Strategy for Targeting Microglia in Development and Disease" for consideration by *eLife*. Your article has been re-reviewed by three peer reviewers, including Isaac M Chiu as the Reviewing Editor and Reviewer #1, and the evaluation has been overseen by Satyajit Rath as the Senior Editor. The following individual involved in review of your submission has agreed to reveal their identity: Amanda Sierra (Reviewer #4).

The reviewers have discussed the reviews with one another and the Reviewing Editor has drafted this decision to help you prepare a revised submission.

Summary:

This study has developed *P2ry12-CreER* mice to genetically mark and target microglia, which are resident immune cells of the CNS. The authors were able to use these mice to distinguish microglia from border associated macrophages, and to track microglia during development and following injury and inflammation. This could be a broadly useful tool for the neuroscience field to study microglial biology and function.

Revisions:

All the reviewers agree that the authors have done a good job addressing the points raised. Major concerns regarding the need for proper cell quantification have been met. Nomenclature issues have been addressed. The Discussion now properly takes into account the comparison with other mouse lines available.

There is only one comment that should be addressed:

Regarding point 13) in their response letter:

They clearly demonstrate that microglia are efficiently (irreversibly) labeled with their strain, and hence previously labeled microglia will remain labeled also in disease models. My earlier question was whether microglia can be labeled/manipulated in an inducible manner once they are reactive i.e. during the course of an inflammatory disease. Given that P2ry12 is downregulated in reactive microglia, the question remains whether levels of Cre are nonetheless sufficient to allow targeting of microglia in inflammation in *P2ry12-CreER* mice. (For example, tamoxifen injection in EAE-diseased animals at peak disease). If they don't have this data, it should at least be discussed.

---

## [Author Response]

Essential revisions:1) Given the leakiness of some CreER strains in combination with certain reporter lines (see Van Hove et al., 2019 and Chappell-Maor et al., 2019), it is critical to analyze the recombination of microglia in P2ry12-CreER Ai14 mice also without tamoxifen treatment.

We thank the reviewer for this suggestion. We performed these experiments and found that there is very low (but not zero) recombination in the absence of tamoxifen using the Ai14 reporter line. We observed a non-tamoxifen recombination rate of 0.38%, or about 1 in every 263 microglia (see Figure 1—figure supplement 1D). Given the sensitivity of the Ai14 reporter allele (Alvarez-Aznar et al., 2020), we believe that this recombination frequency is likely to be lower for other floxed alleles.

2) More rigorous quantification and assessment of recombination in BAMs.Perivascular macrophages can be interrogated in brain sections by IHC, but to achieve a complete assessment of the meningeal BAMs, the meninges need to be removed from inside of the skull. Only the pia layer of the meninges stays adhered to the brain when the skull is removed. LYVE1 is also not an appropriate marker for labeling of BAMs as it only labels a subset. Importantly, dounce homogenization followed by percoll purification is not an appropriate method for isolating BAMs and the expression of TdTomato in other cells than microglia can therefore not be quantified correctly using these approaches.

We very much appreciate these suggestions. We have removed flow cytometry data of macrophages other than microglia (CD11b^+^CD45^int^ cells). Regarding recombination in BAMs, we have generated new pia mater and dura mater whole-mount data from *P2ry12-CreER; Ai14* mice. Labeling these preparations (and thin coronal sections to visualize perivascular macrophages) for two different BAM markers, LYVE1 and CD206, we found that *P2ry12-CreER* does not recombine perivascular or subdural BAMs, but does recombine a subset of CD206+ BAMs in the dura, some of which were also LYVE1+ (Figure 2, Figure 2—figure supplement 2). Our results are consistent with single cell sequencing (Van Hove et al., 2019) which showed low (but not zero) expression of *P2ry12* in meningeal BAMs compared to microglia and CPepi cells (see point 8, below), and with our RiboTrap data which show minimal enrichment of BAM transcripts in immunoprecipitations from *P2ry12-CreER; Rpl22-HA* mice. Importantly, the dura was not separately analyzed in our RiboTrap preparations which would explain why we don’t see enrichment for dural BAM markers in these experiments despite seeing recombination of some dural BAMs in our whole-mount preparations.

3) Effect on P2RY12 protein expression and function.Figure 1A: The authors use single examples of immunostaining against PR2Y12 to demonstrate that the P2ry12-CreER mouse has normal P2RY12 protein expression. However, this claim should be supported by quantification, and immunohistochemistry is not a sensitive method for measuring protein levels. It will be critical to validate that the P2RY12 protein expression in the P2ry12-CreER mice is similar to WT mice with and without tamoxifen treatment. This is straightforward as there are antibodies against P2RY12 commercially available for flow cytometry.

We thank the reviewers for this suggestion. We performed quantitative PCR for *P2ry12* from brains of wild type, heterozygous and homozygous *P2ry12-CreER* mice. We did find a reduction of *P2ry12* mRNA expression in homozygous knock-in mice (please see Figure 1—figure supplement 1A). To test whether this RNA reduction resulted in a reduction in translated protein, we analyzed P2RY12 protein levels in wild-type and homozygous mutant mice, with or without tamoxifen (Figure 1—figure supplement 1B-C). Here, we saw no significant change in P2RY12 protein level in the homozygote mouse, with or without tamoxifen treatment. From this analysis, we conclude that although there is a reduction in *P2ry12* mRNA expression in homozygous knock-in mice, this does not result in a significant change in P2RY12 protein levels.

Additionally, the P2RY12-dependent motility of microglial processes in the P2ry12-CreER mouse should be assessed. Over the last decade, evoking laser-induced injuries has become a standard paradigm in many labs that are studying microglia.

We appreciate this suggestion. Because P2RY12 protein expression is not affected by gene targeting (see above), we believe P2RY12-dependant processes motility is unlikely to be affected in *P2ry12-CreER* mice. However, we did not test process motility directly. While important, we believe that the suggested experiments are beyond the immediate scope of this manuscript – establishing laser-induced injury and live 2-photon microscopy protocols will take several months and require new animal (IACUC) approval.

4) Effect on P2ry12 transcription.Based on the (n=1) ribosomal immunoprecipitations, the p2ry12 transcriptions might appear to be normal and based on this one example; It would be helpful if the authors could quantify P2ry12 transcript levels using qRT-PCR to confirm that the presence of the P2ry12-CreER allele does not impact expression of the gene.

We apologize for the confusion around our ribosome immunoprecipitation protocol. We in fact performed 3 separate immunoprecipitations, each from individual recombined *P2ry12-CreER;Rpl22-HA* mice (3 IPs total, one mouse brain per IP). Three separate libraries were generated from these immunoprecipitations, which were then sequenced along with libraries derived from 3 input controls from the three separate brain lysates prior to immunoprecipitation.

To more quantitatively measure *P2ry12* transcript levels, we performed qPCR analysis as suggested. We observed a reduction in *P2ry12* mRNA in homozygous knock-in mice compared to controls, but no significant difference in P2RY12 protein expression between these same groups (please see Figure 1—figure supplement 1A and response to point #3 above).

5) Reproducibility of Ribosomal immunoprecipitations experiment.This experiment is elegant and support the conclusion that P2ry12-CreER is selective to microglia. However, is this (n=1) experiment reproducible? Please repeat for a minimum of 3 replicate experiments.

We apologize for the confusion here. As described above (point 4), we in fact performed 3 separate immunoprecipitations, each from individual recombined *P2ry12-CreER;Rpl22-HA* mice (3 IPs total, one mouse brain per IP).

It would be very powerful to also perform ribosomal immunoprecipitation on the meninges.

We absolutely agree. As discussed above, we observed *P2ry12-CreER* recombination in LYVE1+ and CD206+ dural macrophages. We would like to transcriptionally profile these cells and compare to *Pf4-Cre* and *Lyve1-Cre* recombined dural, subdural and perivascular macrophages to begin to understand the heterogeneity and transcriptional programs of these BAMs. This will be a focus of future work.

6) In Figure 1B, CD11b and CD45 does not suffice to detect non-parenchymal macrophages by flow cytometry. CD11bhi cells also comprise monocytes, neutrophils, some cDCs and conversely, some borderzone macrophages are CD45low. More markers should be included to identify borderzone macrophages.

Thank you for this suggestion. Our new whole-mount immunohistochemistry and labeling with CD206 and LYVE1 (Figure 2 and Figure 2—figure supplement 2) clearly identifies distinct BAM populations (meningeal – dural/subdural; perivascular; choroid plexus) and shows that *P2ry12-CreER* preferentially targets microglia, and a subset of choroid plexus and dural BAMs, but not other BAM populations. This result is further supported by our ribosome IP sequencing data. We have removed our flow cytometry analysis of non-parenchymal macrophages.

7) In Figure 1C, what is the percentage of microglia, borderzone macrophages and monocytes that are TdTomato+ in P2ry12-CreER Ai14 mice by flow cytometry?

By flow cytometry, the recombination percentage of microglia is 94.8% (Figure 1) and the recombination percentage of circulating monocytes is 0.0175% (Figure 4D). We have removed our flow cytometry analysis of percoll-purified BAMs, but histologically we do not see any recombination in perivascular or pial BAMs (see Figure 2 and Figure 2—figure supplement 2). We do however see some *P2ry12-CreER; Ai14* recombination in dural and choroid plexus macrophages (see Figure 2).

8) In Figure 1E-G, the images suggest that P2ry12-CreER; Ai14 recombination does not occur in perivascular or meningeal macrophages, nor in non-macrophage cells in the brain, but can be detected in choroid plexus macrophages. These conclusions should be supported by quantification, either of immunohistochemistry images, or of FACS data.

We agree with this conclusion, and have added quantification of immunohistochemistry images (please see Figure 2) to provide further support.

In Figure 1G, what are the TdTomato+ cells in P2ry12-CreER; Ai14 mice (Iba1-) in the choroid plexus?

This is a very interesting question. We believe that these cells are CPepi (Kolmer’s epiplexus) cells. We have performed a deeper, more quantitative analysis of *P2ry12-CreER; Ai14* recombination in the adult choroid plexus and have found that all recombined cells are IBA1(+) (see Figure 2). Similar to *Sall1-CreER* (as presented in Van Hove et al., 2019), and in contrast to *Cx3cr1-CreERT2*, which recombines all IBA1+ macrophages in the choroid plexus (see Figure 4—figure supplement 1), we found that *P2ry12-CreER* labeled a subset of IBA1+ CP macrophages on the apical surface of the CP. Van Hove et al. identified several transcripts enriched in putative CPepi-BAMs, including *Clec7a* (also enriched ribosomal IPs from *P2ry12-CreER; Rpl22-HA* mice) and *P2ry12*. Together, these data would suggest that *P2ry12-CreER* recombines CPepi BAMs. It is important to note that in Van Hove et al., 2019, *Sall1-CreERT2* appears to recombine many additional non-macrophage (IBA1-) cells in the wall of the lateral ventricle and choroid plexus stroma (small round cells), whereas *P2ry12-CreER* recombines only IBA1+ cells in the choroid plexus, speaking to the greater specificity of *P2ry12-CreER* recombination.

9) In Figure 2, the authors describe recombination in P2ry12-CreER; Ai14 embryos in microglia, in choroid plexus macrophages and also in some meningeal macrophages. Are perivascular macrophages also labeled?

Yes, in embryos we did see recombination in SMA-adjacent CD206+ macrophages. Please see Figure 3E and discussion below.

Also here, the images should be quantified.

Thank you for this suggestion. We have quantified the % recombination of embryonic meningeal and choroid plexus BAMS in embryos (see Figure 3). As the presence of SMA-coated arteries is very low at the time point we analyzed (E18.5), and the occurrence of CD206+ or LYVE1+ recombined cells is also very low (see Figure 3G), we were unable to quantify perivascular recombination in embryos.

Are macrophages in other embryonic tissues also TdTomato+ or just brain macrophages?

We observed no recombination in embryonic liver or heart. Recombination was low in the embryonic intestine (3.2% of CD206+ cells and 8.2% of IBA1+ cells) and lung (3.2% of CD206+ cells and 5.5% of IBA1+ cells)(see Figure 4—figure supplement 2).

It seems that this CreER strain is not microglia-specific during embryogenesis. This should be discussed in more detail.

Based on our immunostaining, there appears to be a population of cells recombined by *P2ry12-CreER* during embryogenesis that are outside of (but adjacent to) the brain parenchyma. These cells express BAM markers, such as LYVE1 and CD206, and are found in the meningeal space or around aSMA+ vessels (Figure 3D,H). We also see recombination in embryonic IBA1+ cells of the choroid plexus (Figure 3E). Within the brain parenchyma, there is a notable population of *P2ry12-CreER;Ai14* recombined cells that are also CD206+. These cells are most prominent at the junction of the developing dentate gyrus and the meninges overlying the hippocampus (Figure 3G). These CD206+ recombined cells are most concentrated at this position, but form a continuous “stream” through the embryonic subventricular zone (SVZ), although the fraction of CD206+ recombined cells diminishes with greater distance from the hippocampus. Hammond et al. recently described a population of Ms4a-expressing embryonic microglia which co-express both BAM (e.g. *Mrc1*, *CD206*) and microglia (e.g. *P2ry12*, *Fcrls*) transcripts. We speculate that these cells could be described by different models: (1) Embryonic LYVE1+ and CD206+ meningeal macrophages may express *P2ry12* while in border regions, and then die or downregulate BAM genes as they enter the brain parenchyma and become microglia. (2) Some small fraction of cells that will eventually become pial and perivascular macrophages may transiently express *P2ry12* during development and thus be recombined at this time point, (3) These cells may transiently express *P2ry12* while transitioning from microglia to dural BAM state/location, or (4) These cells may express *P2ry12* while in BAM regions and maintain *P2ry12* expression and positioning in these regions (dural BAMs and CPepi cells). The turnover and exchange of BAMs and microglia into and out of their specific niches, and the gene expression programs that these cells adopt, could in theory be quite dynamic. We discuss these findings and models in the Discussion. In the future, we plan to perform more detailed lineage tracing and ablation experiments (beyond the scope of this manuscript) to evaluate these different possibilities.

They should discuss their model also in the context of other previously described models to target microglia (for example Kaiser et al., 2019).

We agree that this is important topic; we have created a separate Discussion section to more explicitly compare these various models.

10) Analyses and quantification of recombination in other organs.In Figure 3, What is the percentage of the macrophages positive for TdTomato in the different organs? This should be quantified (FACS or immunohistochemistry). Where are they located (associated to vessels, peripheral nerves)? The authors state that recombination in tissue-resident macrophages in peripheral organs is “limited” even though they provide images of TdTomato expression in both the spleen, lung, heart, thymus, intestine, and liver. Some level of quantification should be provided to support that this expression is “limited” and negligible. The recombination in the brains of these mice (n = 3 or higher) should also be measured as a positive control to demonstrate high recombination efficiency.

We appreciate this suggestion and have now added quantification of Cx3cr1-EGFP+ and CD206+ recombined cells in various organs from *P2ry12-CreER;Ai14;Cx3cr1GFP* mice. All of these mice had high levels of microglial recombination (over 90%), but recombination in CD206+ or Cx3CR1-EGFP+ cells was low (quantification in Figure 4C). The most significant recombination observed in non-neural tissue was in the marginal zone of the spleen (Figure 4B,C and lower magnification of spleen recombination in images Figure 4—figure supplement 1).

SMA staining in the liver suggests that recombined cells are specifically not associated with SMA+ vessels, unlike *Cx3CR1-EGFP* expressing cells (see Figure 4—figure supplement 1C). In the lung however, recombined cells were specifically associated with SMA-associated airways, or airway adjacent blood vessels (Figure 4—figure supplement 1D). We have not yet had the opportunity to examine whether *P2ry12-CreER; Ai14* recombination is associated with peripheral nerves in these organs.

11) In Figure 3A, there seems to be TdTomato expression in neurons. Can the authors comment on this or provide higher magnification images to exclude co-expression?

Thank you for pointing this out. High magnification images with DAPI show that these are overlapping cells from thick optical sections, and not the same cell (please see new images in Figure 4). Also, please note that neuronal transcripts are specifically depleted in our ribosome IP RNA sequencing data.

12) In Figure 3C, it is not clear why the cells are yellow as there is no GFP.

Thank you for pointing this out. This was a problem with the processing of the image, and this did not reflect actual expression of EGFP. We have corrected the figure.

13) Some microglia signature genes including P2ry12 are downregulated upon microglia activation and in disease. Can this strain be utilized to target activated microglia?

Indeed! Using two different models of injury/inflammation (MCAO stroke and EAE), we have found that *P2ry12-CreER* mice can be used to label what we assume are resident microglia that are responding to injury (inside or near lesion; tdT+P2RY12-TMEM119-) or maintain homeostatic signature (away from lesion; tdT+P2RY12+TMEM119+). We have added these experiments to the manuscript as new Figures 7 and 8, for the MCAO and EAE data respectively.

14) P2ry12 is also highly expressed by platelets. Do the authors find evidence that megakaryocytes are targeted with the P2ry12-CreER;Ai14 mice?

Blood preparations from induced *P2ry12-CreER;Ai14* and *Pf4-Cre;Ai14* mice (well known to recombine platelets) were stained for platelet marker CD41. Although we saw robust labeling of platelets in *PF4-Cre; Ai14* mice, we saw no recombination of platelets in *P2ry12-CreER* mice (see Figure 4). We also found no enrichment for platelet markers in our ribosome sequencing experiments.

15) They show that with Pf4-Cre mice borderzone macrophages can be targeted. This needs better characterization and should be quantified. Are monocytes, tissue macrophages in other organs, or any other cells also labeled?

This mouse line has been well characterized for recombination in monocytes and organ macrophages outside of the CNS (Calaminus et al., 2012, Abram et al., 2014). We have now quantified recombination in BAMs using various markers and whole-mount preparations (Figure 6).

[Editors' note: further revisions were suggested prior to acceptance, as described below.]

Revisions:All the reviewers agree that the authors have done a good job addressing the points raised. Major concerns regarding the need for proper cell quantification have been met. Nomenclature issues have been addressed. The Discussion now properly takes into account the comparison with other mouse lines available.There is only one comment that should be addressed:Regarding point 13) in their response letter:They clearly demonstrate that microglia are efficiently (irreversibly) labeled with their strain, and hence previously labeled microglia will remain labeled also in disease models. My earlier question was whether microglia can be labeled/manipulated in an inducible manner once they are reactive i.e. during the course of an inflammatory disease. Given that P2ry12 is downregulated in reactive microglia, the question remains whether levels of Cre are nonetheless sufficient to allow targeting of microglia in inflammation in P2ry12-CreER mice. (For example, tamoxifen injection in EAE-diseased animals at peak disease). If they don't have this data, it should at least be discussed.

We apologize for not addressing this question in our previous response letter. Whether microglia can be targeted during inflammation is an important question. We suspect that *P2ry12-CreER* expression would be reduced in microglia responding to inflammation and that this would thus reduce the probability of tamoxifen-dependent recombination, but we have not yet tested this. We have added some discussion about this potential strategy in our Discussion section “*P2ry12-CreER* applications”.